# Can you trust your experiments?
# Generalizability of Experimental Studies

## Abstract

Experimental studies are a cornerstone of Machine Learning (ML) research. A common and often implicit assumption is that the study's results will generalize beyond the study itself, e.g., to new data. That is, repeating the same study under different conditions will likely yield similar results. Existing frameworks to measure generalizability, borrowed from the casual inference literature, cannot capture the complexity of the results and the goals of an ML study. The problem of measuring generalizability in the more general ML setting is thus still open, also due to the lack of a mathematical formalization of experimental studies. In this paper, we propose such a formalization, use it to develop a framework to quantify generalizability, and propose an instantiation based on rankings and the Maximum Mean Discrepancy. We show how this latter offers insights into the desirable number of experiments for a study. Finally, we investigate the generalizability of two recently published experimental studies.

## 1 Introduction

Experimental studies are a cornerstone of Machine Learning (ML) research. Due to their importance, the community advocates for high methodological standards when performing, evaluating, and sharing studies (Hothorn et al., 2005; Huppler, 2009; Montgomery, 2017).

The quality of an experimental study depends on multiple independent aspects. First, the experimenter should properly define the *scope* and the *goals* of the study. Particular attention must be given to the choice of benchmarked methods and experimental conditions (Boulesteix et al., 2015; Bouthillier et al., 2021; Dehghani et al., 2021). Second, the study should be *reproducible* by independent parties and hence contain the necessary documentation. This aspect has recently drawn much attention due to the so-called reproducibility crisis (Baker, 2016; Gundersen et al., 2023; Peng, 2011; Raff, 2023; 2021). Third, the results of the study should be sensibly analyzed to draw conclusions regarding, for instance, the *significance* of the findings (Benavoli et al., 2017; Corani et al., 2017; Demsar, 2006). Finally, the *generalizability* of a study concerns how well its results are replicated under unseen experimental conditions, such as datasets not considered in the study (National Academies of Science (2019); Findley et al., 2021; Pineau et al., 2021). The latter two aspects are also known as the internal and external validity of a study. Generalizability and significance, although sometimes confused, are two independent aspects of a study (Findley et al., 2021). On the one hand, significant findings may not generalize to other conditions; on the other hand, results might consistently be not significant.

Generalizability captures how close the results are *between* two different samples of experiments. Generalizability is, conceptually, closely related to model replicability. A model is $\rho$-replicable if, given i.i.d. samples from the same data distribution, the trained models are the same with probability $1 - \rho$ (Impagliazzo et al., 2022). An experimental study is generalizable if, when repeated under different experimental conditions, the results are similar with high probability (National Academies of Science (2019)). A quantifiable notion of generalizability thus requires a formalization of experimental studies, of their results, and of similarity between results.

Significance, instead, captures how strong the findings are *within* the specific sample of experiments performed. Multiple publications have shown how different choices of experimental conditions can lead to very different results (Benavoli et al., 2017; Boulesteix et al., 2017; Bouthillier et al., 2021; Dehghani et al., 2021; Gundersen et al., 2022; Mechelen et al., 2023). Some recent experimental studies have also reported this phenomenon. Matteucci et al. (2023) highlight how previous studies, conducted under different conditions, report different encoders as significantly better than others. Similarly, Lu et al. (2023) re-evaluated coreset learning methods and found that all of the methods they considered did not beat a naïve baseline.

Quantifying generalizability can also help determine the appropriate size of experimental studies. While one dataset is intuitively not enough to draw generalizable conclusions (unless all experiments have the same outcome), $10^6$ datasets likely are. Of course, such large studies are usually not practical: it is crucial to determine the minimum amount of data needed to achieve generalizability. This principle also applies to other experimental factors, such as initialization seed, task, or quality metric.

Our contributions are the following:

1. We introduce a novel measure-theoretic formalization of experimental studies.
2. We propose a quantifiable definition of the generalizability of experimental studies.
3. We develop an algorithm to estimate the size of a study to obtain generalizable results.
4. We analyze two recent experimental studies, Matteucci et al. (2023); Srivastava et al. (2023), and show how well their results generalize.
5. We publish the GENEXPY[1][2] Python module to repeat our analysis in other studies.

Paper outline: Section 2 discusses the related work, Section 3 formalizes experimental studies, Section 4 defines generalizability and provides the algorithm to estimate the required size of a study for generalizability, Section 5 contains the case studies, and Section 6 describes the limitations and concludes.

## 2 Related work

We first discuss the literature related to the problem we are tackling, i.e., why experimental studies may not generalize. Second, we overview the existing concept of model replicability, closely related to our work. Finally, we show other meanings that these words can assume in other domains.

**Non-generalizable results.** It is well known that experimental results can significantly vary based on design choices (Lu et al., 2023; Matteucci et al., 2023; Qin et al., 2023; McElfresh et al., 2022). Possible reasons include an insufficient number of datasets (Dehghani et al., 2021; Matteucci et al., 2023; Alvarez et al., 2022; Boulesteix et al., 2015) as well as differences in hyperparameter tuning (Bouthillier et al., 2021; Matteucci et al., 2023), initialization seed (Gundersen et al., 2023), and hardware (Zhuang et al., 2022). As a result, the statistical benchmarking literature advocates for experimenters to motivate their design choices (Bartz-Beielstein et al., 2020; Mechelen et al., 2023; Boulesteix et al., 2017; Bouthillier et al., 2021; Montgomery, 2017) and clearly state the hypotheses they are attempting to test with their study (Bartz-Beielstein et al., 2020; Moran et al., 2023).

**Replicability and generalizability in ML.** Our work formalizes and extends the definitions of replicability and generalizability given in Pineau et al. (2021) and National Academies of Science (2019). Intuitively, replicable work consists of repeating an experiment on different data, while generalizable work varies other factors as well—e.g., task, seed. In ML, these terms are usually associated to learning algorithms rather than experimental studies. A generalizable model has small generalization error on unseen data (McElfresh et al., 2022), while a replicable model learns the same parameters from different i.i.d. samples (Impagliazzo et al., 2022). Model replicability is also linked to model stability, differential privacy, generalization error, and global stability (Bun et al., 2023; Chase et al., 2023; Ghazi et al., 2023; Moran et al., 2023; Dixon et al., 2023).

---

[1] https://anonymous.4open.science/r/genexpy-B94D
[2] The module will be published on PyPI after acceptance.

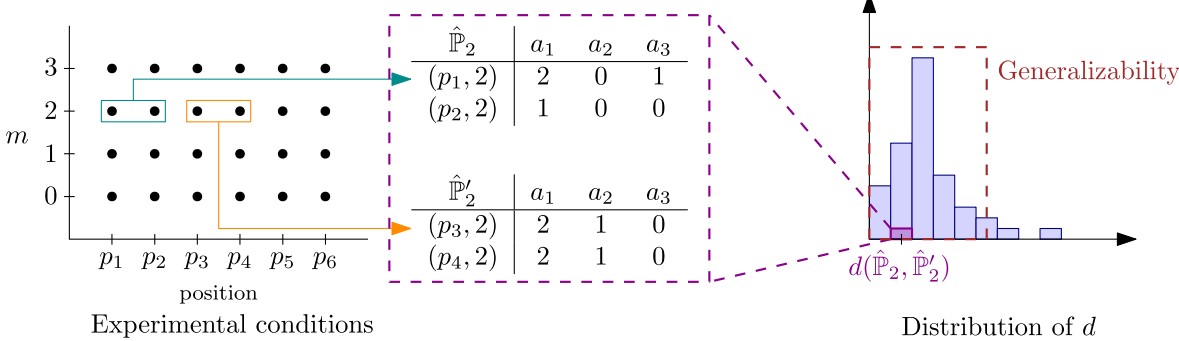

Figure 1: Generalizability of the "checkmate-in-one" task (Example 3.1), as the probability that two sub-studies yield similar results. A result is a distribution of rankings. Note that the design factor ($m$) is fixed, while the generalizability factor (position) varies.

**External validity.** The external validity of a study is a well-studied concept in the context of causal inference, its main applications being in the social and political sciences (Campbell, 1957). In general, the external validity of a study performed concerns whether repeating a study on different samples affects the validity of its findings. Generalizability, opposed to transportability, concerns the external validity of results when the samples come from the same population (Findley et al., 2021). Existing methods assess the sign- and effect-generalization of the treatment on some response variable (Egami & Hartman, 2023). They are thus not applicable to our use-case of ML experimental studies, for which there is—arguably—no treatment and no response variable.

## 3 Experiments and experimental studies

An *experimental study* is a collection of *experiments* comparing the same *alternatives* under different *experimental conditions*. An experimental condition is a tuple of *levels* of *experimental factors*, the parameters defining the experiments. The study aims at answering a *research question*, which defines its *scope* and *goals*.

*Example* 3.1. (The "checkmate-in-one" task, cf. Figure 1) An experimenter wants to compare three Large Language Models (LLMs), the *alternatives*, on the "checkmate-in-one" task (Srivastava et al., 2023; Ammanabrolu et al., 2019; 2020; Dambekodi et al., 2020). The assignment is to find the unique checkmating move from a position of pieces on a chessboard: an LLM succeeds if and only if it outputs the correct move. The experimenter considers two *experimental factors*: the number of shots, $m$, and the initial position on the chessboard, $p_l$. The experimenter wants to find if LLM $a_1$ ranks consistently (in the same position) against the other two LLMs when changing the initial position, for a fixed number of shots.

### 3.1 Experiments

An experiment evaluates all the *alternatives* under a *valid experimental condition*. The *result* of an experiment is an element of some *result space*.

**Alternatives.** An alternative $a \in A$ is an object evaluated in the study, like an LLM in Example 3.1. We call $A$ the finite set of alternatives considered in the study, with cardinality $n_a$.

**Experimental factors.** An experimental factor is *anything* that may affect the result of an experiment. We use $i$ to denote a factor, $C_i$ the (possibly infinite) set of *levels* that $i$ can take, $c \in C_i$ a level of $i$, and $I$ the set of all factors. We adapt Montgomery's classification of experimental factors (Montgomery, 2017, Chapter 1) and distinguish between the *held-constant*, *design*, and *generalizability* factors of a study.

- *Held-constant factors* are presumed not to significantly impact the results, they are hence out of the study's scope and are fixed to a single level; examples are "programming language" or "number of cross-validated folds".
- *Design factors* are expected to significantly impact the results and have a relatively small set of levels; examples are "quality metric" or "number of shots".
- *Generalizability factors* ($I_{\text{gen}}$) have a larger number of levels. The experimenter wants to obtain results that generalize to unseen levels of these factors; examples are "dataset" or "chessboard position".

The same factor may play different roles in different studies, according to the studies' objectives (Montgomery, 2017, Chapter 1). For instance, "seed" is a generalizability factor in reinforcement learning studies such as Nauman et al. (2024), as it typically plays a large role in determining the performance of such algorithms. On the other hand, "seed" is a held-constant factor in Matteucci et al. (2023), as the effect of changing the seed is not of interest for the experimenters.

**Experimental conditions.** An *experimental condition* $\mathbf{c}$ is a tuple of levels of all experimental factors, $\mathbf{c} = (c_i)_{i \in I} \in C \subseteq \prod_{i \in I} C_i$. We assume that all of the $C_i$'s are probability spaces $(i, \mathcal{F}_i, \mu_i)$, and we endow $C$ with the product $\sigma$-algebra $\mathcal{F}_C = \bigotimes_{i \in I} \mathcal{F}_i$ and the product probability measure $\mu^C = \bigotimes_{i \in I} \mu_i$. For instance, if $i$ is "dataset" and $C_i \subseteq \mathbb{R}^{k \times d}$ for some $k, d \in \mathbb{N}$, then $\mu_i$ measures the probability of subsets of $\mathbb{R}^{k \times d}$. If, instead, $i$ is a held-constant factor with $C_i = \{c\}$, then $\mu_i$ is $\mu_i : \{c\} \mapsto 1$. The probability space $(C, \mathcal{F}_C, \mu^C)$ is the *universe of valid experimental conditions*. In our formalization, it plays the role of the sample space for the experiment function $E^Q$ (Section 3.2). Finally, we assume that (1) one can fix the levels of the experimental factors independently, as long as the resulting experimental condition is in $C$; (2) the experimental conditions perfectly model the experiment, i.e., there is a function (the experiment function) mapping a condition into its unique result.

*Example* 3.1 (Continued). $C = \{(p_l, m)\}_{l,m}$, where $p_l$ is a legal configuration of pieces on a chessboard and $m$ is the non-negative number of shots.

**Experimental results.** We define the *result space* as $(\mathcal{X}, \mathcal{B}_{\mathcal{X}})$, where $\mathcal{X}$ is a separable topological space and $\mathcal{B}_{\mathcal{X}}$ is the Borel $\sigma$-algebra on $\mathcal{X}$. For instance, if the experimenter is interested in the raw performances of the alternatives on some learning task, $(\mathcal{X}, \mathcal{B}_{\mathcal{X}}) = (\mathbb{R}^{n_a}, \mathcal{B}_{\mathbb{R}^{n_a}})$.

*Example* 3.1 (Continued). As the goal of the experiment involves ranking the LLMs, the experimenter defines the result of an experiment on $(p_l, m)$ as a ranking of the three LLMs, according to whether or not they output the checkmating move. Suppose that only $a_1$ and $a_2$ output the correct move. Then, the result is $(0, 0, 1)$, where $a_1$ and $a_2$ are tied best.

## 3.2 Experimental studies

A study is defined by its *research question* $Q$, i.e., its *scope* and *goals*. The *scope* consists of the alternatives $A$, the valid experimental conditions $C$, the generalizability factors $I_{\text{gen}}$, and the result space $\mathcal{X}$. The *goal* specifies the kind of conclusion one is attempting to draw from the study. We propose a model of the goals in Section 4.2.

**Definition 3.1** (Research question). The *research question* is a tuple $Q = (A, C, I_{\text{gen}}, \mathcal{X}, \text{goals})$.

*Example* 3.1 (Continued). The research question of the "checkmate-in-one" study is as follows. The *scope* is given by $A = \{a_1, a_2, a_3\}$, $C = \{(p_l, m)\}_{l,m}$, $I_{\text{gen}} = \{\text{"position"}\}$, and $\mathcal{X}$ the set of rankings of 3 alternatives. The *goal* is *"Is $a_1$ consistently in the same position of the ranking?"*

To model the results, we introduce the *experiment function* $E^Q : (C, \mathcal{F}_C, \mu^C) \rightarrow (\mathcal{X}, \mathcal{B}_{\mathcal{X}})$, which evaluates all of the alternatives in $A$ under a valid experimental condition. We also assume that $E^Q$ is a measurable function or, equivalently, a random variable with sample space $C$. We can thus sample from $E^Q$.

**Result of an experiment.** The *result of the experiment performed under experimental condition* $\mathbf{c}$ *is* $E^Q(\mathbf{c})$. The *experiment* is the act of applying $E^Q$ to an experimental condition. For instance, let's say an experimenter is ranking ML classifiers under multiple datasets, with cross-validation. If "dataset" is the only experimental factor and $\mathbf{c} = d$, then $E^Q(d)$ ranks the models according to their average cross-validated performance on dataset $d$. If, instead, "fold" is also a factor, $\mathbf{c} = (d, f)$, then $E^Q((d, f))$ ranks the models according to their performance on each fold independently. .

**Result of a study.** The *result of the study on a research question* $Q$ is the distribution of $E^Q$: since $E^Q$ is not necessarily injective, we assign higher probability to those results that appear more often.

**Definition 3.2** (Result of a study)**.** The *result of the study on* $Q$ is the pushforward probability induced by the experiment function $E^Q$,

$$\mathbb{P}^Q := E^Q_* \mu^C : \mathcal{B}_{\mathcal{X}} \to [0, 1]$$
$$Y \mapsto \mu^C \left( E^{Q^{-1}}(Y) \right),$$

where $E^{Q^{-1}}(Y) = \left\{ \mathbf{c} : E^Q(\mathbf{c}) \in Y \right\} \in \mathcal{F}_C$ is the preimage of $Y$.

Intuitively, the result $\mathbb{P}^Q$ is a function mapping a measurable subset $Y$ of $\mathcal{X}$ to the measure of the sets of experimental conditions having $Y$ as result. In practice, as $C$ might be infinite or too large, one can only run experiments under $n$ experimental conditions and obtain a sample $\hat{E}^Q_n \sim \mathbb{P}^Q$ of size $n$. We call this an *empirical study of size $n$ on $Q$* and denote the empirical distribution of $\hat{E}^Q_n$ with $\hat{\mathbb{P}}^Q_n$. To keep the notation clean and unless necessary, we omit $Q$.

## 4 Generalizability of experimental studies

The currently accepted definition of generalizability is the property of two independent studies with the same research question to yield similar results, see National Academies of Science (2019) and Pineau et al. (2021). Although intuitive, this notion is not practically useful as it cannot be assessed objectively. We thus propose the following quantifiable definition of generalizability based on our framework (cf. Section 3.1)

**Definition 4.1** (Generalizability)**.** Let $Q = (A, C, I_{\text{gen}}, \mathcal{X}, \text{goals})$ be a research question, let $\mathbb{P}$ be the result of the corresponding study, and let $d$ be a distance between probability distributions. The *$n$-generalizability of the study on $Q$* is

$$n\text{-Gen}(Q; \varepsilon) := \Pr_{X, Y \sim \mathbb{P}^n} (d(X, Y) \le \varepsilon) \tag{1}$$

where $\varepsilon \in \mathbb{R}^+$ is a dissimilarity threshold. With an abuse of notation, we can have $d$ accept samples as inputs, in which case it compares their empirical distributions.

Intuitively, the $n$-generalizability is the probability, computed with resampling, for any two empirical studies of size $n$ on $Q$ to yield "similar" results, as defined by $d$ and $\varepsilon$. We discuss in Section 4.4 how to interpretably choose a value for $\varepsilon$.

Definition 4.1 is very flexible and allows for different choices of the result space $\mathcal{X}$, the goals, and the distance $d$. The rest of this section proposes a particular instantiation, based on the Maximum Mean Discrepancy (Gretton et al., 2006), which allows for different choices of the experimental space $\mathcal{X}$ and goals.

### 4.1 Experimental results: Rankings with ties

Experimental results can be formalized in different ways, such as raw performance metrics, time series, or rankings. Among these, rankings are arguably one of the most natural forms:

(i) Rankings are already widely used for non-parametric tests such as Friedman, Nemenyi, and Conover-Iman (Demsar, 2006; Conover & Iman, 1982).

(ii) Rankings do not suffer from experimental-condition-fixed effects, such as a dataset being inherently easier to solve than another one. Even though there are multiple ways to deal with these effects, there

is no preferred one in the literature. See, for instance, the consensus ranking problem (Matteucci et al., 2023; Nießl et al., 2022).

(iii) Rankings allow the definition of interpretable kernels to formalize different goals of a study, as we illustrate in Section 4.2.

We define rankings (with ties) in the following way.

**Definition 4.2** (Ranking). A ranking $r$ on $A$ is a transitive and reflexive binary endorelation on $A$. Equivalently, $r$ is a totally ordered partition of $A$ into *tiers* of equivalent alternatives. $r(a)$ denotes the *rank* of $a \in A$, i.e., the position of the tier of $a$ in the ordering. W.l.o.g., we use $\mathcal{R}_{n_a}$ for the space of rankings of $n_a$ alternatives.

## 4.2 Goals: Kernels

Goals act as lenses, focusing on specific aspects of the results that are of interest for the experimenter. For instance, consider the goal in Example 3.1: *"Is $a_1$ consistently in the same position of the ranking?"*. In this case, the goal solely focuses on $a_1$'s position in the rankings, ignoring the positions of $a_2$ and $a_3$. Changing the goal of a study can thus heavily impact how the results are analyzed. Within our framework, we formalize for goals as *kernels* on the result space, i.e., positive definite symmetric functions. We choose kernels as our instantiation of distance, and thus of $n$-generalizability, depends on the MMD. As we will discuss in the next Section, however, not every distance supports kernels or, in general, ways to include the goal of a study. In the following, we describe three kernels for rankings, covering three representative goals.

**Borda kernel.** The Borda kernel is suitable for goals in the form *"Is alternative $a^*$ consistently ranked the same?"*. It uses the Borda count, defined as the number of alternatives (weakly) dominated by a given one (Borda, 1781). For a pair of rankings, we compute the Borda counts of $a^*$ and then take their difference.

$$\kappa_b^{a^*, \nu}(r_1, r_2) = e^{-\nu |b_1 - b_2|},$$

where $b_l = |\{a \in A : r_l(a) \geq r_l(a^*)\}|$ is the number of alternatives dominated by $a^*$ in $r_l$ and $\nu \in \mathbb{R}^+$. The Borda kernel takes values in $[e^{-\nu n_a}, 1]$. If $\nu$ is too large compared to $1/|b_1 - b_2|$, the kernel is oversensitive and will heavily penalize even small differences. On the contrary, if $\nu$ is too small, the kernel is undersensitive and will not penalize deviations unless they are very large. As $|b_1 - b_2| \in [0, n_a]$, we recommend $\nu = 1/n_a$.

**Jaccard kernel.** The Jaccard kernel is suitable for goals in the form *"Are the best alternatives consistently the same ones?"*. As it measures the similarity between sets (Gärtner et al., 2006; Bouchard et al., 2013), we use it to compare the top-$k$ tiers of two rankings.

$$\kappa_j^k(r_1, r_2) = \frac{\left| r_1^{-1}([k]) \cap r_2^{-1}([k]) \right|}{\left| r_1^{-1}([k]) \cup r_2^{-1}([k]) \right|},$$

where $r^{-1}([k]) = \{a \in A : r(a) \leq k\}$ is the set of alternatives with rank lower than or equal to $k$. The Jaccard kernel takes values in $[0, 1]$.

**Mallows kernel.** The Mallows kernel is suitable for goals in the form *"Are the alternatives ranked consistently?"*. It measures the overall similarity between rankings (Jiao & Vert, 2018; Mania et al., 2018; Mallows, 1957). We adapt the original definition in (Mallows, 1957) for ties,

$$\kappa_m^\nu(r_1, r_2) = e^{-\nu n_d},$$

where $n_d = \sum_{a_1, a_2 \in A} |\text{sign}(r_1(a_1) - r_1(a_2)) - \text{sign}(r_2(a_1) - r_2(a_2))|$ is the number of discordant pairs and $\nu \in \mathbb{R}^+$. If a pair is tied in one ranking but not in the other, one counts it as half a discordant pair. The Mallows kernel takes values in $\left[\exp\left(-2\nu\binom{n_a}{2}\right), 1\right]$. If $\nu$ is too large compared to $1/n_d$, the kernel is oversensitive and it will heavily penalize even small differences. On the contrary, if $\nu$ is too small, the kernel is undersensitive and will not penalize deviations unless they are very large. As $n_d \in \left[0, \binom{n_a}{2}\right]$, we recommend $\nu = 1/\binom{n_a}{2}$.

The following example illustrates the three kernels.

*Example* 4.1. Consider two rankings $\mathbf{r} = (0, 0, 0)$ and $\mathbf{s} = (0, 1, 1)$, where $x_j$ is the rank of the $j$-th alternative. In $\mathbf{r}$, all three alternatives are tied best in tier 0, while in $\mathbf{s}$ $a_1$ is the best (in tier 0): $a_2$ and $a_3$ are tied worst in tier 1. To understand their impact on generalizability, consider a study whose result is a distribution assigning both $\mathbf{r}$ and $\mathbf{s}$ probability $1/2$. For the goal corresponding to the Borda kernel, $\mathbf{r}$ and $\mathbf{s}$ answer the research question consistently as $a_1$ weakly dominates all alternatives in both rankings. Hence, the Borda kernel takes a value of 1 and the study is perfectly generalizable. For the Jaccard and Mallows goals, instead, the two rankings are either very different ($\kappa_j^1(r_1, r_2) \approx 0.33$) or slightly different ($\kappa_m^{1/n_a^2}(r_1, r_2) \approx 0.72$). Thus, we conclude that the study is more generalizable w.r.t. the Mallows kernel than the Jaccard kernel.

### 4.3 Distance between experimental results: Maximum Mean Discrepancy

In the previous sections, we have formalized an experimental study, its results, and its goals. The last open point before applying (1) in practice is a definition of $d$, a distance between probability distributions. Such a distance should satisfy the following requirements. First, it should take into consideration the goal of a study. Second, it should handle sparse distributions well: empirical studies are typically very small compared to the number of all possible rankings, which grows super-exponentially in the number of alternatives.[3] Third, it should provide a way to indicate the amount of experiments needed to achieve $n$-generalizable results. A distance satisfying the above requirements is the Maximum Mean Discrepancy (MMD) (Gretton et al., 2006; 2012; Mania et al., 2018). Other related approaches, such as Rastogi et al. (2022) for rankings, are not able to model the different goals a study may have.

**Definition 4.3** (MMD). Let $\mathcal{X}$ be a set with a kernel $\kappa$, and let $\mathbb{Q}_1$ and $\mathbb{Q}_2$ be two probability distributions on $\mathcal{X}$. Let $\mathbf{x} = (x_i)_{i=1}^n \sim \mathbb{Q}_1, \mathbf{y} = (y_i)_{i=1}^m \sim \mathbb{Q}_2$.

$$\text{MMD}(\mathbf{x}, \mathbf{y})^2 := \frac{1}{n^2} \sum_{i,j=1}^n \kappa(x_i, x_j) + \frac{1}{m^2} \sum_{i,j=1}^m \kappa(y_i, y_j) - \frac{2}{mn} \sum_{\substack{i=1\ldots n \\ j=1\ldots m}} \kappa(x_i, y_j).$$

Finally, there remains the choice of an appropriate $\varepsilon^*$ to use in (1), which is hardly interpretable. The following result relates the range of the MMD and the infimum and supremum of the kernel.

**Proposition 4.1.** *The MMD takes values in* $\left[0, \sqrt{2 \cdot (\kappa_{sup} - \kappa_{inf})}\right]$, *where* $\kappa_{sup} = \sup_{x,y \in X} \kappa(x, y)$ *and* $\kappa_{inf} = \inf_{x,y \in X} \kappa(x, y)$.

Using a similar approach, we propose to replace $\varepsilon^*$ with a condition on the desired minimum expected value of the kernel:

$$\varepsilon^*(\delta^*) = \sqrt{2\left(\kappa_{\sup} - f_\kappa(\delta^*)\right)}, \tag{2}$$

where $f_\kappa$ is a kernel-specific function and $\delta^*$ is an interpretable parameter. We now discuss them for the three kernels discussed in Section 4.2 with their recommended parameters.

- *Borda kernel.* $\delta^*$ is $|b_1 - b_2|/n_a$, i.e., the difference between the fraction of dominated alternatives between two rankings; $f_{\kappa_b}(x) = e^{-x}$.
- *Jaccard kernel.* $\delta^*$ is the Jaccard coefficient between the top-$k$ tiers of two rankings; $f_{\kappa_j}(x) = 1 - x$.
- *Mallows kernel.* $\delta^*$ is the fraction of discordant pairs; $f_{\kappa_m}(x) = e^{-x}$.

As a concrete example, achieving ($\alpha^* = 0.90, \delta^* = 0.05$)-generalizable results for the Jaccard kernel means that, w.p. 0.90, the average Jaccard coefficient between two rankings drawn from the results is at least 0.95.

### 4.4 An estimate for the necessary number of experiments

When designing a study, the experimenter has to decide how many experiments to run in order to obtain generalizable results. In other words, they need to choose a (minimum) sample size $n^*$ that achieves the

---

[3]Fubini or ordered Bell numbers, `https://oeis.org/A000670`.

desired generalizability $\alpha^*$ for a given threshold $\varepsilon^*$.

$$n^* = \min \left\{ n \in \mathbb{N} : n\text{-Gen}\,(Q; \varepsilon^*) \geq \alpha^* \right\}. \tag{3}$$

*Example* 3.1 (Continued). The experimenter wants to obtain results in which, with probability 0.99, $a_1$ dominates the same number of alternatives up to a difference of 1. This does *not* happen, for instance, if $a_1$ dominates all 3 alternatives in one ranking and just 1 (itself) in another one. They therefore choose the Borda kernel with $\nu = 1/n_a$, $\delta^* = 1/3$, and $\varepsilon^* = \sqrt{2}\sqrt{1 - e^{-0.33}}$ as in (2). *How many experiments are enough?*

Let now $\varepsilon_n^{\alpha^*}$ be the $\alpha^*$-quantile of the MMD. To estimate $n^*$, we use a linear dependency between $\log(n)$ and $\log(\varepsilon_n^{\alpha^*})$, which we have observed in our experiments; see, for instance, Figure 2. In particular, for any distribution $\mathbb{P}$ and choice of $\alpha^* \in [0,1]$, there exist $\beta_0, \beta_1 \in \mathbb{R}$, $\beta_i = \beta_i(\alpha^*, \mathbb{P})$, s.t.

$$\log(n) = \beta_1 \log\left(\varepsilon_n^{\alpha^*}\right) + \beta_0. \tag{4}$$

Appendix A.3.1 provides a proof for the distribution-free case. In practice, however, estimates of $n^*$ made with the distribution-free bound are over conservative, so we stick to the empirical formula. Equation (4) suggests that one can use a small set of $N$ preliminary experiments to estimate $n^*$, iteratively improving the estimate by considering more experimental conditions. In practice, however, sampling a random experimental condition often relies on a pre-fixed pool. This reflects the reality of many experimental studies, where the experimenter defines the factors and their possible levels upfront. On this result is based Algorithm 1, whose working is illustrated in Figure 2.

---

**Algorithm 1** Estimate the necessary number of experiments

---

**Require:** $\alpha^*$        ▷ desired generalizability
**Require:** $\delta^*$        ▷ similarity threshold on rankings
**Require:** $Q$        ▷ research question, $Q = (A, C, I_{\text{gen}}, \kappa)$
**Require:** $N$        ▷ number of initial experiments
**Require:** $N_{\text{step}}$        ▷ number of additional experiments
**Require:** $n_{\text{rep}}$        ▷ number of repetitions to estimate the distribution of the MMD

  **procedure** ESTIMATENSTAR$(\alpha^*, \delta^*, Q, N, n_{\text{max}}, n_{\text{rep}})$
    $\hat{n}_N^* \leftarrow \infty$
    **while** $N < \hat{n}_N^*$ **do**
      Run $N$ experiments and get their results $\hat{\mathbb{P}}_N$
      $\varepsilon^* \leftarrow f_\kappa(\delta^*)$        ▷ cf. Section 4.4
      $n_{\text{max}} \leftarrow \lfloor N/2 \rfloor$        ▷ we need two disjoint samples of size $n_{\text{max}}$ from $\hat{\mathbb{P}}_N$
      **for** $n = 1 \dots n_{\text{max}}$ **do**
        mmds $\leftarrow$ empty list
        **for** $n = 1 \dots n_{\text{rep}}$ **do**
          sample without replacement $(x_j)_{j=1}^{2n} \sim \hat{\mathbb{P}}_N$
          $\mathbf{x} \leftarrow (x_j)_{j=1}^{n}$        ▷ split into disjoint samples
          $\mathbf{y} \leftarrow (x_j)_{j=n+1}^{2n}$
          append MMD $(\mathbf{x}, \mathbf{y})$ to mmds
        **end for**
        $\varepsilon_n^{\alpha^*} \leftarrow \alpha^*$-quantile of mmds
      **end for**
      fit a linear regression $\log(n) = \beta_1 \log\left(\varepsilon_n^{\alpha^*}\right) + \beta_0$
      $\hat{n}_N^* \leftarrow \beta_1 \log(\varepsilon^*) + \beta_0$
      $N \leftarrow N + N_{\text{step}}$
    **end while**
    **return** $n_N^*$
  **end procedure**

---

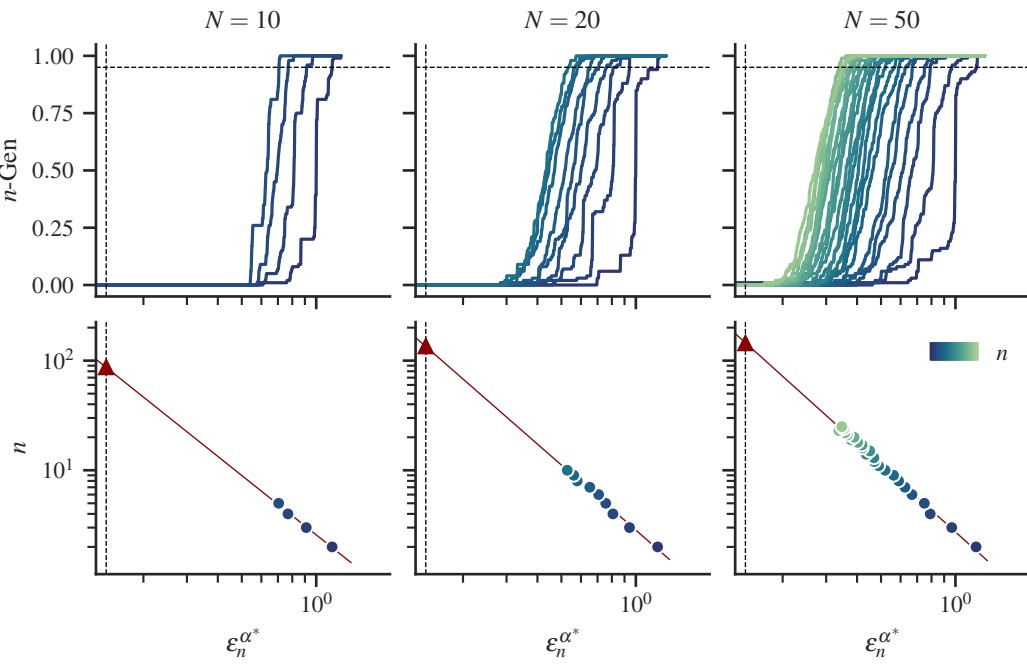

Figure 2: Estimation of the desired number of experiments $\hat{n}_N^*$ (marked with ▲) from different amounts $N$ of preliminary experiments and $n \in \{1, \dots, \lfloor N/2 \rfloor\}$. The dashed lines represent $\alpha^* = 0.9$ (horizontal) and $\varepsilon^* = \sqrt{2}/10$ (vertical). We used real data from Matteucci et al. (2023) and the Jaccard kernel (Section 4.2).

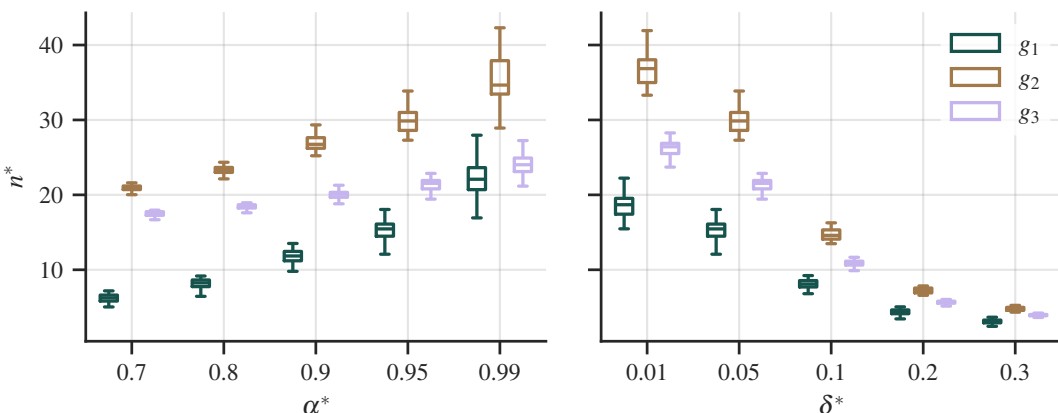

Figure 3: Number of necessary experiments $n^*$ to achieve generalizability for categorical encoders, for different desired generalizability $\alpha^*$, similarity threshold $\delta^*$, goals $g_i$. The variation in the plot is due to the combinations of design factors.

## 5 Case studies

### 5.1 Case Study 1: A benchmark of categorical encoders

We now evaluate the generalizability of a recent study (Matteucci et al., 2023) that analyzes the performance of encoders for categorical data. The performance of an encoder is approximated by the quality of a model trained on the encoded data. The *design factors* are "model", "tuning strategy" for the pipeline, and "quality metric" for the model, while the only *generalizability factor* is "dataset". We impute missing values in the results of the study by assigning the worst rank. We evaluate how well the results of the study generalize w.r.t. three goals:

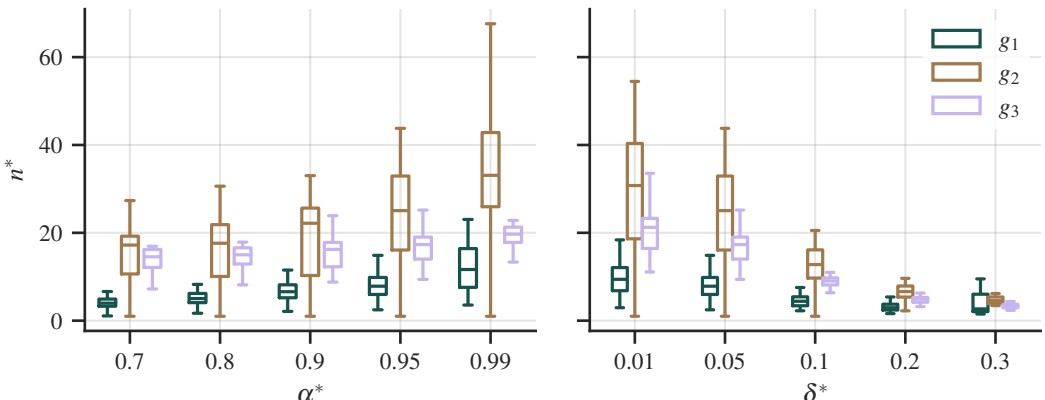

Figure 4: Number of necessary experiments $n^*$ to achieve generalizability for LLMs, for different desired generalizability $\alpha^*$, similarity threshold $\delta^*$, goals $g_i$. The variation in the plot is due to the combinations of design factors.

($g_1$) Find out if the one-hot encoder (a popular encoder) ranks consistently amongst its competitors, using the Borda kernel with $\nu = 1/n_a$.

($g_2$) Investigate if some encoders outperform all the others using the Jaccard kernel with $k = 1$.

($g_3$) Evaluate whether the encoders are ranked in a similar order, using the Mallows kernel with $\nu = 1/\binom{n_a}{2}$.

Figure 3 shows the predicted $n^*$ for different choices of $\alpha^*$ and $\delta^*$, the other one fixed at 0.95 and 0.05 respectively. The variance in the boxes comes from variance in the design factors. For example, the results for the design factors "decision tree, full tuning, accuracy" have a different $(\alpha^*, \delta^*)$-generalizability than the results for "SVM, no tuning, accuracy". We observe on the left that—as expected—obtaining generalizable results requires more experiments as the desired generalizability $\alpha^*$ increases. We can also see that the variance of the boxes increases with $\alpha^*$, meaning that the choice of the design factors has a larger influence on the achieved generalizability. This is also possibly due to tail effects in the estimation of the MMD. We observe the same when decreasing $\delta^*$, as it corresponds to a stricter similarity condition on the rankings. In the rather extreme cases of $\alpha^* = 0.7$ or $\delta^* = 0.3$, even less than 10 datasets are enough to achieve $(\alpha^*, \delta^*)$-generalizability.

Consider now goal $g_2$ for two different choices of design factors: (A): "decision tree, full tuning, accuracy" and (B): "SVM, full tuning, balanced accuracy". Furthermore, let $(\alpha^*, \delta^*) = (0.95, 0.05)$: we estimate $n^* = 28$ for (A) and $n^* = 34$ for (B), corresponding to the bottom and top whiskers of the corresponding box in Figure 3. As both (A) and (B) were evaluated using $n = 30$ experiments, we conclude that the results of (A) are (barely) $(0.95, 0.05)$-generalizable, while those of (B) are not. Hence, one should run more experiments with fixed factors (B) to make the study generalizable.

## 5.2 Case study 2: BIG-bench — A benchmark of Large Language Models

We now evaluate the generalizability of BIG-bench (Srivastava et al., 2023), a collaborative benchmark of Large Language Models (LLMs). The benchmark compares LLMs on different tasks, such as the "checkmate-in-one" task (cf. Example 3.1), and for different numbers of shots. "Task" and "number of shots" are the *design factors*. Every task has a number of "subtasks", which is the *generalizability factor*. We stick to the preferred scoring for each subtask. As the results have too many missing values to impute them, we only consider the experimental conditions where at least 80% of the LLMs had results, and to the LLMs whose results cover at least 80% of the conditions.

As before, we define three goals:

($g_1$) Find out if GPT3 (to date, one of the most popular LLMs) ranks consistently amongst its competitors, using the Borda kernel with $\nu = 1/n_a$.

($g_2$) Investigate if some encoders outperform all the others using the Jaccard kernel with $k = 1$.

($g_3$) Evaluate whether the LLMs are ranked in a similar order, using the Mallows kernel with $\nu = 1/\binom{n_a}{2}$.

Figure 4 shows the predicted $n^*$ for different choices of $\alpha^*$ and $\delta^*$, the other one fixed at 0.95 and 0.05 respectively. Again, the variance in the boxes comes from variance in the design factors, i.e., the task and the number of shots. As before, increasing $\alpha^*$ or decreasing $\delta^*$ leads to higher $n^*$. Unlike in the previous section, $n^*$ for $g_2$ greatly depends on the combination of fixed factors, as we now detail.

Consider now goal $g_2$ for two different choices of design factors: (A): "conlang_translation, 0 shots", and (B): "arithmetic, 2 shots". Furthermore, let $(\alpha^*, \delta^*) = (0.95, 0.05)$. For this choice of parameters, we estimate $n^* = 44$ for (A), corresponding to the top whisker of the corresponding box in Figure 3. As the study evaluates (A) on 10 subtasks, it is therefore not $(0.95, 0.05)$-generalizable. In fact, we estimate that this would require 34 more subtasks. For (B), on the other hand, we estimate $n^* = 1$: the best 2-shot LLM for the observed subtasks is, for all of the 21 subtasks, PALM 535B. Now, if any of the 44 LLMs were equally likely to outperform the others, this would happen w.p. $1/44^{21} \approx 3e - 35$. However, we argue that this is not the case. In the context of our framework, we can instead say that the true results show that PALM 535B is always the best one. At least, for the estimate of the true results that we obtain from running the experiments on the 21 subtasks. It is therefore not unlikely for PALM 535B to outperform the others, rather, it's certain. As the result of every single experiment is the same, even a study performed on one experiment is perfectly $(0.95, 0.05)$-generalizable—in fact, it is $(1, 0)$-generalizable. In other words, our algorithm was able to quantify *in hindsight* that a single experiment would have been enough to obtain generalizable results. Of course, however, one cannot trust an estimate of $n^*$ based on only one experiment. The next section thus investigates how the number of preliminary experiments influences the estimate of $n^*$.

### 5.3 How many preliminary experiments?

We assess the accuracy of our method (cf. Section 4.4) for estimating $n^*$ from $N$ independent experiments. Our procedure is as follows. First, we selected $\alpha^* = 0.95$ and $\delta^* = 0.05$. Second, we select a probability distribution $\mathbb{P}$ on the set of rankings $\mathcal{R}_{n_a}$, representing the true distribution of outcomes. Third, for various values of $n$, we bootstrap the distribution of $\text{MMD}(\mathbb{P}^n, \mathbb{P}^n)$ by repeatedly sampling independently from $\mathbb{P}$. We compute $n^*$ as in (3), i.e., as the minimum $n$ guaranteeing $(\alpha^*, \delta^*)$-generalizability. Using synthetic distributions allows us to sample as many results as needed. To evaluate our estimate, we generate $N \in 10, 20, 40, 80$ samples and we estimate $n^*$ from the empirical distribution of results $\hat{\mathbb{P}}_N$, using Algorithm 1. We call the estimate $\hat{n}^*_N$. This procedure is repeated across multiple distributions, with 100 repetitions per distribution.

Figure 5 displays the relative error $(\hat{n}^*_N - n^*)/n^*$: when greater (lower) than zero, it indicates the overestimation (underestimation) of $n^*$. Although the specific results vary with the target objective (with goal $g_2$, corresponding to the Jaccard kernel, being particularly challenging to estimate), in general, $\hat{n}^*_N$ approximates $n^*$ within 50% in more than 75% of cases, even when $N = 10$. Consequently, our method provides a reliable estimate of $n^*$ (or at least of its order of magnitude) from as few as 10 preliminary experiments. Appendix B.1 contains additional details regarding the distributions and the results, as well as an application to real data.

## 6 Conclusion

**Limitations.** We only investigated the instantiation of our framework based on the MMD, rankings, and kernels for rankings. Keeping the MMD as the cornerstone of an instantiation allows different result spaces and goals while potentially maintaining crucial properties such as (4). In this regard, a close formula for the coefficients in (4) would greatly benefit the computational efficiency and the theoretical understanding of our framework.

**Future work.** First, incorporating information about the experimental conditions into the framework, for instance, by including information about the datasets. This would allow to study other aspects of external validity, such as transportability, as well as allow for active learning to choose the next experiments to run.

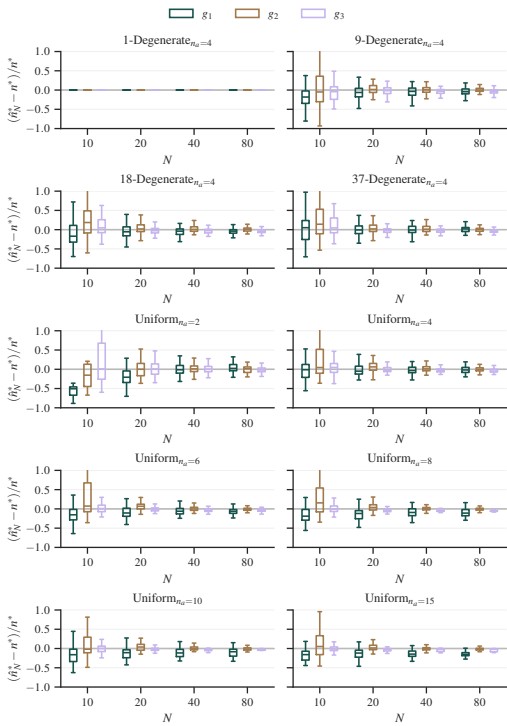

Figure 5: Relative error of the prediction of $n^*$ from $N$ preliminary experiments $(n_N^*)$ for the goals $g_i$.

Second, based on our experiments in Section 5.3, we intend to provide guarantees and confidence intervals on the convergence of $n_N^*$ to $n^*$. Third, we dealt with missing evaluations by imputing them. Having kernels that can handle missing evaluations might be beneficial. Fourth, rankings, despite their advantages, do not consider the raw performance difference between alternatives. On the other hand, kernels for raw performances (i.e., for vectors in $\mathbb{R}^{n_a}$) lack an obvious interpretation as the goals of a study. Fuzzy rankings may bridge this gap: performance differences are incorporated into the ranking and the existing kernels for rankings might be adapted to them. Additionally, our framework assumes that all alternatives are evaluated from all studies and under all conditions. Moving to partial rankings might solve the limitation. Finally, our framework might help develop tools to identify cherry-picked result. For instance, one can isolate outliers by comparing the results of multiple studies in a meta-analysis fashion.

**Conclusions.** An experimental study is generalizable if, with high probability, its findings will hold under different experimental conditions, e.g., with unseen datasets. Non-generalizable studies might be of limited use or even misleading. This paper is, to our knowledge, the first to develop a quantifiable notion for the generalizability of experimental studies. To achieve this, we formalize experiments, experimental studies, and their results, as well as define an instantiation—rankings and distributions over rankings. Our approach allows us to estimate the number of experiments needed to achieve a desired level of generalizability in new experimental studies. We demonstrate its utility showing generalizable and non-generalizable results in two recent experimental studies.

**Acknowledgments**

. . .

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

# A  Details for Section 4

## A.1  Details for Section 4.2

This section contains the proofs to show that the functions introduced in Section 4.2 are kernels, i.e., symmetric and positive definite. As symmetry is a clear property of all of them, we only discuss their positive definiteness. Our proofs for the Borda and Mallows kernels follow Jiao & Vert (2018): we define a distance $d$ on the set of rankings $\mathcal{R}_{n_a}$ and show that $(\mathcal{R}_{n_a}, d)$ is isometric to an $L_2$ space. This ensures that $d$ is a conditionally positive definite (c.p.d.) function and, thus, that $e^{-\nu d}$ is positive definite (Schoenberg, 1938; Schölkopf, 2000). Our proof for the Jaccard kernel, instead, follows without much effort from previous results. For ease of reading, we restate the definitions as well.

**Definition A.1** (Borda kernel).

$$\kappa_b^{a^*, \nu}(r_1, r_2) = e^{-\nu|b_1 - b_2|}, \tag{5}$$

where $b_l = \#\{a \in A : r_l(a) \geq r_l(a^*)\}$ is the number of alternatives dominated by $a^*$ in $r_l$ and $\nu \in \mathbb{R}$.

**Proposition A.1.** *The Borda kernel as defined in* (5) *is a kernel.*

*Proof.* Define a distance

$$
\begin{aligned}
d : \mathcal{R}_{n_a} \times \mathcal{R}_{n_a} &\to \mathbb{R}^+ \\
(r_1, r_2) &\mapsto \|b_1 - b_2\|,
\end{aligned}
$$

where $b_l = \{a \in A : r_l(a) \geq r_l(a^*)\}$ is the number of alternatives dominated by $a^*$ in $r_l$. Now, $(\mathcal{R}_{n_a}, d)$ is isometric to $(\mathbb{R}, \|\cdot\|_2)$ via the map $r_l \mapsto b_l$. Hence, $d$ is c.p.d. and $\kappa_b$ is a kernel. □

**Definition A.2** (Jaccard kernel).

$$\kappa_j^k(r_1, r_2) = \frac{\left| r_1^{-1}([k]) \cap r_2^{-1}([k]) \right|}{\left| r_1^{-1}([k]) \cup r_2^{-1}([k]) \right|}, \tag{6}$$

where $r^{-1}([k]) = \{a \in A : r(a) \leq k\}$ is the set of alternatives whose rank is better than or equal to $k$.

**Proposition A.2.** *The Jaccard kernel as defined in* (6) *is a kernel.*

*Proof.* It is already know that the Jaccard coefficients for sets is a kernel (Gärtner et al., 2006; Bouchard et al., 2013). As the Jaccard kernel for rankings is equivalent to the Jaccard coefficient for the $k$-best tiers of said rankings, the former is also a kernel. □

**Definition A.3** (Mallows kernel).

$$\kappa_m^\nu(r_1, r_2) = e^{-\nu n_d}, \tag{7}$$

where $n_d = \sum_{a_1, a_2 \in A} |\mathrm{sign}(r_1(a_1) - r_1(a_2)) - \mathrm{sign}(r_2(a_1) - r_2(a_2))|$ is the number of discordant pairs and $\nu \in \mathbb{R}$.

**Proposition A.3.** *The Mallows kernel as defined in* (7) *is a kernel.*

*Proof.* The number of discordant pairs $n_d$ is a distance on $\mathcal{R}_{n_a}$ (Snell & Kemeny, 1962). Consider now the mapping of a ranking into its adjacency matrix,

$$
\begin{aligned}
\Phi : \mathcal{R}_{n_a} &\to \{0, 1\}^{n_a \times n_a} \\
r &\mapsto (\mathbf{1}(r(i) \leq r(j)))_{i,j=1}^{n_a},
\end{aligned}
$$

where $\mathbf{1}$ is the indicator function. Then,

$$n_d = \|\Phi(r_1) - \Phi(r_2)\|_1 = \|\Phi(r_1) - \Phi(r_2)\|_2^2$$

where $\|\cdot\|_p$ indicates the entry-wise matrix $p$-norm and the equality holds because the entries of the matrices are either 0 or 1. As a consequence, $(\mathcal{R}_{n_a}, n_d)$ is isometric to $(\mathbb{R}^{n_a \times n_a}, \|\cdot\|_2)$ via $\Phi$. Hence, $n_d$ is c.p.d. and $\kappa_m$ is a kernel. □

## A.2 Details for Section 4.3

**Proposition 4.1.** *The MMD takes values in* $\left[0, \sqrt{2 \cdot (\kappa_{sup} - \kappa_{inf})}\right]$, *where* $\kappa_{sup} = \sup_{x,y \in X} \kappa(x,y)$ *and* $\kappa_{inf} = \inf_{x,y \in X} \kappa(x,y)$.

*Proof.*

$$0 \leq \mathrm{MMD}_\kappa \left(\mathbf{x}, \mathbf{y}\right)^2 = \frac{1}{n^2} \sum_{i,j=1}^{n} \kappa(x_i, x_j) + \frac{1}{m^2} \sum_{i,j=1}^{m} \kappa(y_i, y_j) - \frac{2}{mn} \sum_{\substack{i=1\ldots n \\ j=1\ldots m}} \kappa(x_i, y_j)$$

$$\leq \frac{1}{n^2} \sum_{i,j=1}^{n} \kappa_{\mathrm{sup}} + \frac{1}{m^2} \sum_{i,j=1}^{n} \kappa_{\mathrm{sup}} - \frac{2}{mn} \sum_{\substack{i=1\ldots n \\ j=1\ldots m}} \kappa_{\mathrm{inf}} = 2(\kappa_{\mathrm{sup}} - \kappa_{\mathrm{inf}})$$

□

## A.3 Details for Section 4.4

### A.3.1 Distribution-free linear relation

**Proposition A.4.** *Let* $\varepsilon_n^\alpha$ *be the* $\alpha$-*quantile of the MMD for samples of size* $n$. *For any* $\alpha \in [0,1]$, *there exist* $\beta_0, \beta_1 \in \mathbb{R}$, $\beta_i = \beta_i(\alpha)$ *and* $\overline{\varepsilon_n^\alpha} \geq \varepsilon_n^\alpha$, *s.t., for any distribution,*

$$\log(n) = \beta_1 \log\left(\overline{\varepsilon_n^\alpha}\right) + \beta_0. \tag{8}$$

*Proof.* The proof goes as follows. First, we find a close formula for $\overline{\varepsilon_n^\alpha}$ using Gretton et al. (2012, Theorem 8). Then. we show the linear dependency in (8).

Let $X$ and $Y$ be iid samples of size $n$ from an arbitrary distribution $\mathbb{P}$ and define the random variable $\mathrm{MMD}_n = \mathrm{MMD}(X, Y)$.

$$\mathbb{P}^n \otimes \mathbb{P}^n \left(\mathrm{MMD}_n - \sqrt{\frac{2\kappa_{\mathrm{sup}}}{n}} > \varepsilon\right) < \exp\left(-\frac{n\varepsilon^2}{4\kappa_{\mathrm{sup}}}\right)$$

$$\overset{(1)}{\Longrightarrow} \mathbb{P}^n \otimes \mathbb{P}^n \left(\mathrm{MMD}_n > \varepsilon'\right) < \exp\left(-\frac{n\left(\varepsilon' - \sqrt{\frac{2\kappa_{\mathrm{sup}}}{n}}\right)^2}{4\kappa_{\mathrm{sup}}}\right)$$

$$\overset{(2)}{\Longrightarrow} \mathbb{P}^n \otimes \mathbb{P}^n \left(\mathrm{MMD}_n > n^{-\frac{1}{2}}\left(\sqrt{-\log(1-\alpha)\,4\kappa_{\mathrm{sup}}} + \sqrt{2\kappa_{\mathrm{sup}}}\right)\right) < 1 - \alpha$$

$$\overset{(3)}{\Longrightarrow} \mathbb{P}^n \otimes \mathbb{P}^n \left(\mathrm{MMD}_n \leq n^{-\frac{1}{2}}\left(\sqrt{-\log(1-\alpha)\,4\kappa_{\mathrm{sup}}} + \sqrt{2\kappa_{\mathrm{sup}}}\right)\right) \geq \alpha \tag{9}$$

where:

(1) Define $\varepsilon' = \varepsilon + \sqrt{2\kappa_{\mathrm{sup}}/n}$.

(2) Define $1 - \alpha = \exp\left(-\frac{n\left(\varepsilon' - \left(\frac{2\kappa_{\mathrm{sup}}}{n}\right)\right)^2}{4\kappa_{\mathrm{sup}}}\right)$ and $\varepsilon' = n^{-\frac{1}{2}}\left(\sqrt{-\log(1-\alpha)\,4\kappa_{\mathrm{sup}}} + \sqrt{2\kappa_{\mathrm{sup}}}\right)$.

(3) Take the complementary event.

By definition of $\alpha$-quantile and (9), it is clear that $\overline{\varepsilon_n^\alpha} \geq \varepsilon_n^\alpha$.

Now, define

$$\overline{\varepsilon_n^\alpha} := n^{-\frac{1}{2}} \left( \sqrt{-\log\left(1-\alpha\right)4\kappa_{\mathrm{sup}}} \right) + \sqrt{2\kappa_{\mathrm{sup}}}.$$

From (9), it follows that

$$n = \left(\overline{\varepsilon_n^\alpha}\right)^{-2} \left( \sqrt{-4\kappa_{\mathrm{sup}}\log\left(1-\alpha\right)} + \sqrt{2\kappa_{\mathrm{sup}}} \right)^2$$

and, taking logarithms, that

$$\log(n) = -2\log(\overline{\varepsilon_n^\alpha}) + 2\log\left( \sqrt{-4\kappa_{\mathrm{sup}}\log\left(1-\alpha\right)} + \sqrt{2\kappa_{\mathrm{sup}}} \right).$$

Defining

$$\begin{aligned} \beta_0 &= \log\left(2\kappa_{\mathrm{sup}}\right) + \log\left( \sqrt{-2\log\left(1-\alpha\right)} + 1 \right) \\ \beta_1 &= -2 \end{aligned} \tag{10}$$

concludes the proof. $\qquad\square$

*Remark.* Proposition A.4 can be used to obtain a close-formula estimate for $n^*$, using $\beta_0$ and $\beta_1$ as in (10). However, we have observed that this leads to overconservative estimates compared to the method described in Section 4.4 (usually one or two orders of magnitude of difference). Hence, we recommend using Algorithm 1.

## B  Details for Section 5

### B.1  Estimation of $n^*$ from $N$ preliminary experiments

In this section, we discuss how accurately our method estimates $n^*$ from $N$ independent experiments (cf. Section 4.4). Recall that, for a research question $Q$ with corresponding results $\mathbb{P}$,

$$n^* = \min\left\{ n \in \mathbb{N} : \Pr_{X,Y\sim\mathbb{P}^n}\left(\mathrm{MMD}(X,Y) \leq \varepsilon\right) \geq \alpha^* \right\},$$

where $\alpha^*$ and $\varepsilon^*$ are the desired generalizability and dissimilarity respectively. We estimate $n^*$ by running $N$ preliminary experiments and using their results $\hat{\mathbb{P}}_N$ as an approximation for $\mathbb{P}$ (cf. Algorithm 1). We call the obtained estimate $\hat{n}_N^*$.

We evaluate our approach with the relative error $(n^* - \hat{n}_N^*)/n^*$. A relative error greater (lower) than 0 means that we are overestimating (underestimating) $n^*$.

#### B.1.1  Synthetic data

We experimented with two families of distributions.

**Uniform**$_{n_a}$  The uniform distribution assigns the same probability to every ranking in $\mathcal{R}_{n_a}$. In our experiments, we varied $n_a$ from 2 to 15. Perhaps surprisingly, our estimate improved with the number of alternatives, despite the distribution becoming more "complex".

$m$-**Degenerate**$_{n_a}$  The degenerate distribution assigns the same probability to $m$ rankings in $\mathcal{R}_{n_a}$. As $m$ approaches $|\mathcal{R}_{n_a}|$, the $m$-degenerate distribution becomes closer to a uniform. In our experiments, we fixed

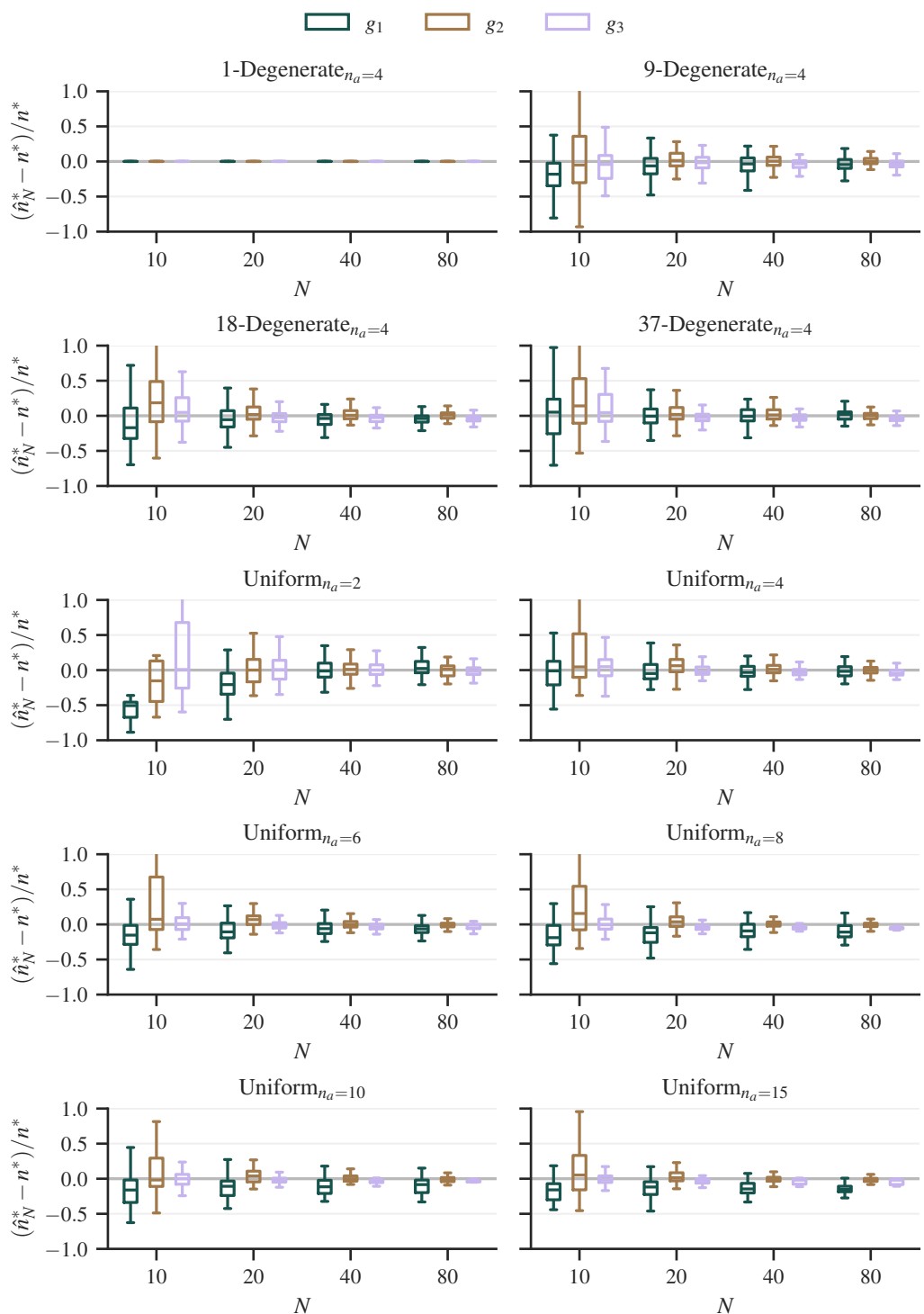

Figure 6: Relative error of the prediction of $n^*$ from $N$ preliminary experiments $(\hat{n}_N^*)$ for multiple synthetic distributions and the three kernels introduced in Section 4.2.

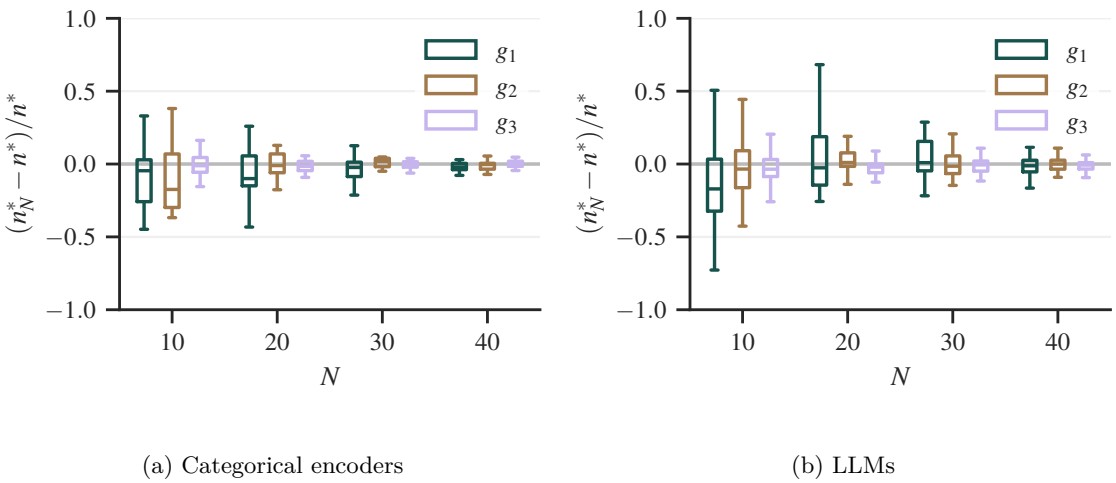

(a) Categorical encoders

(b) LLMs

Figure 7: Relative error between the estimate of $n^*$ from $N$ preliminary experiments and $n_{50}^*$.

$n_a = 4$ ($|\mathcal{R}_{n_a}| = 75$) and experimented with $m$ ranging from $^{75}/_8 \approx 9$ to $^{75}/_2 \approx 37$. This parameter doesn't seem to affect the results.

Figure 6 shows that there are no major discrepancies in the behavior of our estimate.

### B.1.2 Real data

This section evaluates the influence of the number of preliminary experiments $N$ on $n^*$. We consider, for both case studies (cf. Section 5), the design factor combinations for which we have at least 50 experiments. This results in 23 out of 48 combinations for the categorical encoders and 9 out of 24 combinations for the LLMs. For each of those combinations and as we do not have access to the true value $n^*$, we consider the estimate $n_{50}^*$ as the ground truth and observe how the estimates of $n^*$ for $N < 50$ differ. Figure 7 shows that even $N = 10$ preliminary experiments provide an estimate within of 50% of $n_{50}^*$, validating the fact that our method can return reasonably good estimates even for very small sample size.

