# OpenReview forum: "Can you trust your experiments? Generalizability of Experimental Studies"
_TMLR — Rejected by TMLR_

### Review · Reviewer_CgtL · 2025-04-30

**Summary Of Contributions:**

The authors propose a formalization of generalizability in machine learning experiments. In particular: the authors define:
- a research question $Q$ is a tuple $Q=(A, C, I_{gen}, \mathcal{X}, \text{goals})$, where
	- $A$ is a set of alternatives (e.g. competing models)
	- $I_{gen}$ is a set of experimental factors that the experimenter wants to generalize over (e.g. datasets)
	- $C$ is a set of experimental conditions (e.g. set of datasets, task settings) with associated measure $\mu^C$
	- $\mathcal{X}$ is the result space, with an associated $\sigma$-algebra $\mathcal{B}_{\mathcal{X}}$
	- goals: the conclusion attempted to be drawn e.g. ``is $a_1$ consistently in the same position in the ranking''

- an experiment function $E^Q: (C, \mathcal{F}_C, \mu^C) \to (\mathcal{X}, \mathcal{B}_{\mathcal{X}})$, a random variable with sample space $C$.

- the result of a study is the pushforward probability induced by $E^Q$, that is, $\mathbb{P}^Q \eqdef E^Q \mu^C: \mathcal{B}_{\mathcal{X}} \to [0,1]$.

The authors then define generalizability as the probability that two empirical studies of size $n$ on $Q$ are $\epsilon$-close in some distance metric $d$.

For distance metrics, the authors propose a number of ranking-based options.

The authors consider two studies to illustrate their method.

**Audience:**

No

**Claims And Evidence:**

No

**Requested Changes:**

Questions

- What does this mean "Matteucci et al. (2023) discuss how previous studies on categorical encoders disagree on the
best-performing ones, even when the results are significant." How are the results significant when they disagree? Do the authors mean: within a study, a certain categorical encoder was reported to be significantly better than others (based on the mean and standard error of a metric), but in other studies, other categorical encoders were better?

- What is $\mathcal{F}_C$?

- It seems like $A$ is missing in the definition of the experiment function - where does $A$ come in? What if there is randomness in $A$ - for example, the same model gives different results for the same experimental factor $c$? How is this captured in the model?

- Can you clarify the notation
$ Y\mapsto \mu^C({E^{Q}}^{-1}(Y))$: What is the function that maps $Y$ to the output? Can you explain what $Y$ is?

- Definition 4.1. Can you explain this definition:
$\mathbb{P}^n \otimes \mathbb{P}^n((X_j)_{j=1}^n, (Y_j)_{j=1}^n: d(X, Y) \leq \epsilon))$? In particular:
- what is $Y_j$? Is $Y_j$ the same distribution as $X_j$? For example, is it results on the same dataset? Results from a different dataset? What is the randomness here, intuitively? Is it from randomness in the algorithms $A$, or from random seeds?

- Example 4.1: can you clarify these rankings? In $r=(0,0,0)$ does this mean there were three models being compared, and they all were assigned the same tier, $0$? In $s=(0,1,1)$, the first model was in tier 0, and the other two were in tier 2?

- Example 4.1: ``Hence, the Borda kernel takes a value of 1 and the study is perfectly generalizable." This seems too strong a statement - generalizable over what? There is no mention of sample size etc. Also isn't this probabilistic? There may be a case where $a_1$ does not dominate.

- Equation 2: why is this a good choice for $\epsilon^*$? Don't you want the MMD to be as small as possible? Why are you using the maximum value of the MMD as the choice for $\epsilon*$? Won't it always be less than the maximum, so the generalization is vacuous?

- p. 7 "Borda kernel: $\delta^*$ is the difference between the fraction of dominated alternatives between two rankings". This is not clear to me. What is the "fraction" referring to?

- "How many experiments to ensure generalizability?" - this language also seems too strong to me - I don't think you can guarantee/ensure generalizability - this is still probabilistic.

- p. 7 "$a_1$ dominates the same number of alternatives up to a difference of 1". This sentence is unclear. Does it mean $a_1$ dominates the alternatives in all experiment conditions except for one condition?

- Eq. 4: the authors suppose a linear relationship between $log(n)$ and $log(\epsilon_n^{\alpha^*})$. They provide a proof in Appendix A.3.1 - however, I'm not sure the proof is valid -  it seems like $\alpha$ still depends on $n$. They also say that the estimate can be orders of magnitude off.

- In Algorithm 1, I'm confused where the sample size of each experiment comes in. If you have 100 experiments, but each have a sample size of 10, there is so much variability in the metric that you may not be able to draw any conclusions from any of the experiments - you would need to aggregate them somehow.

- Figure 2: what is the scale of $n$ in the legend? I find it difficult to read this plot.

- Section 5.1. I am also confused again by what is meant by experiment in this case. Do you mean new data? Do you mean evaluating the results again on the same data with different random seeds? If it is new data, what are the properties of the new data? Is it similarly distributed to the previous data?

- Section 5.2. I'm confused by $n^*$ vs. $n^*_N$? What are the different ways they are being predicted?

- "For (B), on the other hand, we estimate $n^*=1$": the best 2-shot LLM for the observed subtasks is always PALM 535B. Hence the result of a single experiment is enough to acheive (0.95, 0.05)-generalizability". Can the authors explain a little more here? Is the idea: there are $T$ subtasks, all of which PALM performs best on, which would be extremely unlikely to happen by chance? In this case, shouldn't $T$ be an important variable to note?

**Strengths And Weaknesses:**

## Strengths

I agree that generalizability in ML is an important problem, and that more systematic comparison of methods and experimental conditions are warranted.


## Weaknesses

I find the notation quite difficult to parse. I also find the section 4.4 difficult to understand - that is, the steps to determine the number of experiments to run. Due to this, it's difficult to map the concepts in the paper back to empirical studies in ML (see questions below for more details).

I'm also not convinced we need this complexity. For example, suppose I consider 3 models on $T$ tasks. Model 1 is the best on all $T$ tasks. I can construct a hypothesis test to see if this is an "extreme" result, indicating model 1 is not just peforming better by chance. (E.g. if we compare on 1 task, there is 33% chance model 1 is the best under this null hypothesis). Specifically: $H_0$: model 1 does not dominate models 2, 3 vs. $H_1$: model 1 dominates models 2,3. To obtain a null distribution, I can sample ranks uniformly $T$ times (representing $T$ tasks). Then, I compare my observed data to this test statistic. I can also compute the power of this test numerically to determine how many tasks I need for a particular power level.

Philosophically, I also disagree with the authors that the way to improve generalizability of ML experiments is to provide a measure-theoretic formulation of generalizability. Instead, I think that a higher level of disclosure is necessary e.g. providing code base, including random seeds and hyperparameter tuning details, and the full set of datasets considered (as opposed to cherry-picked datasets for the paper). However, I acknowledge this is my personal opinion and taste.

---

> ### Author Response · Authors · 2025-05-26
>
> We wish to thank the reviewer for their useful feedback and comments. We have revised the manuscript, marking in _orange_ the related changes.
>
> - **I find the notation quite difficult to parse. I also find the section 4.4 difficult to understand - that is, the steps to determine the number of experiments to run. Due to this, it's difficult to map the concepts in the paper back to empirical studies in ML (see questions below for more details).**
>
> We thank the reviewer for this feedback. We would appreciate clarification on which aspects of the notation were particularly difficult to parse, as this would help us improve the clarity of the manuscript. Regarding Section 4.4, we have revised the text to more clearly describe the step-by-step process for estimating the number of experiments required, and we’ve added additional commentary to better connect these steps to common empirical workflows in machine learning. We hope these changes make the procedure more accessible and welcome any further suggestions for clarification.
>
> - **I'm also not convinced we need this complexity. For example, suppose I consider 3 models on T tasks. Model 1 is the best on all T tasks. I can construct a hypothesis test to see if this is an "extreme" result, indicating model 1 is not just peforming better by chance. (E.g. if we compare on 1 task, there is 33% chance model 1 is the best under this null hypothesis). Specifically: H0: model 1 does not dominate models 2, 3 vs. H1: model 1 dominates models 2,3. To obtain a null distribution, I can sample ranks uniformly T times (representing T tasks). Then, I compare my observed data to this test statistic. I can also compute the power of this test numerically to determine how many tasks I need for a particular power level.**
>
> We thank the reviewer for the thoughtful example and acknowledge that traditional hypothesis testing, as described, is effective for assessing _significance_, that is, the strength or extremeness of results within a given sample of tasks. However, our aim is fundamentally different: we focus on _generalizability_, which concerns whether the results observed on one sample of tasks will resemble those obtained on a new, independently drawn sample. As discussed in the introduction, while significance addresses *internal validity* (i.e., are the results meaningful within the current study?), generalizability addresses *external validity* (i.e., will these results likely hold under new experimental conditions?). The added complexity in our approach stems from this broader objective. Rather than assessing if a model is significantly better within a fixed sample, we estimate how many experimental conditions are needed to ensure that the findings are stable and repeatable across studies. This distinction is critical for evaluating the robustness of empirical claims in machine learning, particularly as experimental settings become increasingly diverse and high-dimensional.
>
> - **Philosophically, I also disagree with the authors that the way to improve generalizability of ML experiments is to provide a measure-theoretic formulation of generalizability. Instead, I think that a higher level of disclosure is necessary e.g. providing code base, including random seeds and hyperparameter tuning details, and the full set of datasets considered (as opposed to cherry-picked datasets for the paper). However, I acknowledge this is my personal opinion and taste.**
>
> We appreciate the reviewer’s perspective and fully agree that transparency, through practices such as sharing code, random seeds, hyperparameter details, and complete datasets, is essential for improving _reproducibility_. These practices allow others to precisely replicate and verify a study's findings.
> However, our work addresses a different concern: _generalizability_, which pertains to whether a study’s conclusions are likely to hold under new experimental conditions. While reproducibility ensures that a study can be re-run as originally conducted, it does not guarantee that the results will generalize beyond that specific setting.
> Our formalization is intended to provide a principled, quantitative way to assess generalizability, which we see as complementary to, rather than in conflict with, efforts to improve reproducibility. Both are crucial to the robustness and reliability of empirical research in machine learning.
>
> **What does this mean "Matteucci et al. (2023) discuss how previous studies on categorical encoders disagree on the best-performing ones, even when the results are significant." How are the results significant when they disagree? Do the authors mean: within a study, a certain categorical encoder was reported to be significantly better than others (based on the mean and standard error of a metric), but in other studies, other categorical encoders were better?**
>
> Thank you for raising this point. We have clarified the sentence.

---

> > ### Author Response · Authors · 2025-05-26
> >
> > - **What is FC?**
> >
> > We thank the reviewer for raising this point. Previously, $\mathcal F_C$ was implicitly defined as the product $\sigma$-algebra on $C$, formally needed to define $\mu^C$ but otherwise not mentioned elsewhere. We clarified the meaning of $\mathcal F_C$ in Section 3.1, § Experimental conditions.
> >
> > - **It seems like A is missing in the definition of the experiment function - where does A come in? What if there is randomness in A - for example, the same model gives different results for the same experimental factor c? How is this captured in the model?**
> >
> > The experiment function $E^Q$, as defined in our notation, depends on the research question $Q$, which explicitly includes the set of alternatives $ A $, the valid experimental conditions $ C $, the generalizability factors $I_\text{gen}$, and the result space $ X $ (see Definition 3.1). Thus, $ A $ is inherently part of the function.
> > Regarding randomness in $A$ (for example, when comparing non-deterministic methods) this variability is captured by introducing another factor, such as "seed", so that the experiment function remains deterministic. Especially in ML, this is a realistic assumption as standard computers are deterministic.
> > Currently, our framework assumes the set of alternatives is fixed across studies, allowing comparison only when they share the same alternatives (or their intersection). We acknowledge that extending the model to handle varying alternatives through partial rankings and suitable kernels is a valuable direction and have included this as future work.
> >
> > - **Can you clarify the notation Y↦μC(EQ−1(Y)): What is the function that maps Y to the output? Can you explain what Y is?**
> >
> > In Definition 3.2, we use standard mathematical notation to define the function $\mathbb P^Q$, which is the pushforward probability measure induced by the experiment function $E^Q$. This function $\mathbb P^Q$ maps any measurable subset $Y$ of the result space $\mathcal X$ (i.e., any element of the Borel sigma-algebra $B_\mathcal X$) to the probability assigned to that set.
> > Formally, $Y$ represents a measurable subset of possible experimental results, and $\mathbb P^Q(Y) = \mu^C({E^Q}^{-1}(Y))$ gives the measure of the set of all experimental conditions  whose results lie within $Y$. This notation is standard in probability theory when defining pushforward measures.
> >
> > - **Definition 4.1. Can you explain this definition: $\mathbb{P}^n \otimes \mathbb{P}^n((X_j)*{j=1}^n, (Y_j)*{j=1}^n: d(X, Y) \leq \epsilon))$**? **In particular: what is Yj? Is Yj the same distribution as Xj? For example, is it results on the same dataset? Results from a different dataset? What is the randomness here, intuitively? Is it from randomness in the algorithms A, or from random seeds?**
> >
> > We thank the reviewer for these questions and have updated Definition 4.1 and the *Experimental Conditions* section to clarify the underlying assumptions and notation.
> > In the expression above,  the variables $X = (X_j)_j$ and $Y = (Y_j)_j$ are two independent samples of size $n$, each drawn from the result distribution $\mathbb{P} = \mathbb{P}^Q$ defined in Definition 3.2. This models the scenario of repeating the same study twice—under independently sampled experimental conditions—and comparing the empirical results.
> > To clarify:
> > - $X_j$ and $Y_j$ are results (e.g., rankings) from two independently sampled experimental conditions.
> >     - They are drawn from the same distribution $\mathbb{P}$, which captures the variability over all valid experimental conditions for a fixed set of alternatives.
> >     - The randomness in $\mathbb{P}$ comes from the variation in experimental conditions—for example, datasets, hyperparameters, or other controlled factors.
> >     - We assume alternatives (e.g., algorithms or models) are fixed. Any internal randomness in these alternatives (e.g., non-deterministic training procedures) can be modeled by including the random seed as part of the experimental conditions.
> >
> > We hope these revisions clarify that generalizability, as we define it, measures how stable the empirical results are across different sets of experimental conditions—not across different alternatives or runs of a non-deterministic model.
> > Finally, we updated the formula and text in definition 4.1 to make it easier to read.

---

> > > ### Author Response · Authors · 2025-05-26
> > >
> > > - **Example 4.1: can you clarify these rankings? In r=(0,0,0) does this mean there were three models being compared, and they all were assigned the same tier, 0? In s=(0,1,1), the first model was in tier 0, and the other two were in tier 2?**
> > >
> > > The reviewer’s interpretation is correct. In Example 4.1, the ranking $r = (0, 0, 0)$ indicates that three models were compared and all were assigned to the same tier (0) implying they are equivalent in performance. In contrast, the ranking $s = (0, 1, 1)$ means the first model was placed in tier 0 (best), while the second and third models were grouped together in tier 1, indicating a lower but equal performance relative to the first. We added this explanation to Example 4.1.
> > >
> > > - **Equation 2: why is this a good choice for ϵ∗? Don't you want the MMD to be as small as possible? Why are you using the maximum value of the MMD as the choice for ϵ∗? Won't it always be less than the maximum, so the generalization is vacuous?**
> > >
> > > We thank the reviewer for this question and appreciate the opportunity to clarify. We are not proposing to set $\varepsilon^\*$ equal to the maximum possible value of the MMD. As the reviewer rightly points out, doing so would trivially result in perfect generalizability for any study.
> > > Instead, Equation (2) presents a parametrized formulation of $\varepsilon^\*$ that is *inspired* by the range of the MMD given in Proposition 4.1. Specifically, we avoid relying on the loose lower bound $\kappa_\inf{}$ and instead introduce a kernel-dependent function $f_\kappa(\delta^\*)$, where $\delta^\*$ is a user-defined, interpretable parameter that expresses a desired minimum expected similarity under the kernel. Equation (2) thus defines a non-trivial threshold on the MMD that reflects how similar we desire two sets of results to be, on average, under the chosen kernel.
> > >
> > > The choice of $\delta^\*$ (and thus $\varepsilon^\*$) is ultimately left to the user, and we explore the impact of different values in our case studies in Section 5. This flexibility allows practitioners to adjust the strictness of the generalizability criterion to suit their specific domain or application.
> > >
> > >
> > >
> > > - **p. 7 "Borda kernel: δ∗ is the difference between the fraction of dominated alternatives between two rankings". This is not clear to me. What is the "fraction" referring to?**
> > >
> > > We thank the reviewer for pointing out the ambiguity. In the context of the Borda kernel, each alternative is assigned a score equal to the number of alternatives it dominates in a given ranking. If $b$ denotes this number, then the fraction of dominated alternatives is defined as $b / n_a$, where $n_a$ is the total number of alternatives.
> > > This fraction provides a normalized measure of an alternative’s relative standing in the ranking. We clarified this point in Section 4.3.
> > >
> > > - **"How many experiments to ensure generalizability?" - this language also seems too strong to me - I don't think you can guarantee/ensure generalizability - this is still probabilistic.**
> > >
> > > We understand the reviewer's criticism and replaced the title with a weaker formulation.
> > >
> > > - **p. 7 "a1 dominates the same number of alternatives up to a difference of 1". This sentence is unclear. Does it mean a1 dominates the alternatives in all experiment conditions except for one condition?**
> > >
> > > We appreciate the reviewer’s request for clarification. The statement refers to a comparison within two individual results (rankings), not across multiple experimental conditions. Specifically, it means that the number of alternatives dominated by $a_1$ differs, on average, by at most 1 between two rankings. For example, consider the rankings $r = (0, 0, 0)$ and $s = (1, 1, 0)$. In $r$, $a_1$ is tied with all alternatives and thus dominates all 3. In $s$, $a_1$ dominates 2 alternatives ($a_1$ and $a_2$). The difference in the number of alternatives dominated is 1, satisfying the condition. We added this example to Example 4.1.

---

> > > > ### Author Response · Authors · 2025-05-26
> > > >
> > > > - **Eq. 4: the authors suppose a linear relationship between log(n) and log(ϵnα∗). They provide a proof in Appendix A.3.1 - however, I'm not sure the proof is valid - it seems like α still depends on n. They also say that the estimate can be orders of magnitude off.**
> > > >
> > > > We thank the reviewer for their close reading and comments on Equation (4) and the proof in Appendix A.3.1.
> > > > First, we clarify that the theoretical, distribution-free estimate of $n^*$ discussed in Appendix A.4 is *not* used in our experiments, nor do we propose it as a practical method. As noted in the appendix, this estimate tends to be overly conservative and is included primarily for completeness.
> > > >
> > > > Regarding the proof itself: we acknowledge that the relationship among $\log(n)$, $\log(\varepsilon_n^\alpha)$, and $\alpha$ may appear circular. However, this is due to the fact that these three quantities are intrinsically linked, as $\varepsilon_n^\alpha$ denotes the $\alpha$-quantile of the distribution of $\mathrm{MMD}_n$. Hence, by definition, $\alpha$, $n$, and $\varepsilon_n^\alpha$ are not independent: the dependency of $\alpha$ on $n$ arises naturally because we are describing the $\alpha$-quantile of a distribution that itself depends on the sample size $n$.
> > > >
> > > > To address this concern and improve clarity, we revised the proof of Proposition A.4 and the accompanying remark to better explain the role of each quantity and how the distribution-free approximation is justified. We hope this resolves the confusion and would be happy to receive more specific feedback if further clarification is needed.
> > > >
> > > > - **In Algorithm 1, I'm confused where the sample size of each experiment comes in. If you have 100 experiments, but each have a sample size of 10, there is so much variability in the metric that you may not be able to draw any conclusions from any of the experiments - you would need to aggregate them somehow.**
> > > >
> > > > We thank the reviewer for raising this important point. As defined in Section 3.2, an experiment in our framework yields a single result. The concept of sample size $n$ applies at the level of the study, which is a collection of $n$ experiments conducted under independently sampled experimental conditions. This is the sample size that appears in Algorithm 1.
> > > >
> > > > If the reviewer is referring to cases where each experiment involves repeated evaluations (e.g., due to cross-validation or stochastic model training), then those repetitions would indeed need to be aggregated (e.g., via averaging) to yield a single result (e.g., ranking) per experiment. Our framework assumes this aggregation is done *before* modeling, so that each experiment contributes one well-defined result to the overall study. Alternatively, if one wishes to preserve variability across repeated evaluations (e.g., across cross-validation folds), this could be modeled by introducing an additional experimental factor such as `"cv_fold"`. Whether including such variability explicitly would affect generalizability remains an open question. We updated Section 3.2 to include this clarification.
> > > >
> > > > - **Figure 2: what is the scale of n in the legend? I find it difficult to read this plot.**
> > > >
> > > > We thank the reviewer for this observation. Figure 2 is intended to illustrate the iterative process of Algorithm 1. We begin with $N = 10$ and incrementally increase it as needed. At each step, we estimate the $n$-generalizability for $n \in \{1, \dots, \lfloor N/2\rfloor\}$, fit the linear model described in Equation (4), and obtain an estimate of $\hat n^\*_N$. If $N < \hat n^\*_N$​, more experiments are sampled and the process repeats. We have updated the figure caption to explicitly describe the scale and range of $n$.

---

> > > > > ### Author Response · Authors · 2025-05-26
> > > > >
> > > > > - **Section 5.1. I am also confused again by what is meant by experiment in this case. Do you mean new data? Do you mean evaluating the results again on the same data with different random seeds? If it is new data, what are the properties of the new data? Is it similarly distributed to the previous data?**
> > > > >
> > > > > We thank the reviewer for this question. Formally, an experiment is the application of the experimental function to a chosen experimental condition. Informally, it corresponds to selecting a specific configuration (i.e., a setting of the experimental factors) and running the procedure needed to produce a result in the target format. In the specific context of Section 5.1, each experimental condition is a tuple (model, tuning strategy, quality metric, dataset). The original study does not report investigating random seeds, so we do not treat "seed" as an experimental factor. Running an experiment thus involves selecting a new combination of these four components and evaluating the encoders accordingly. We then rank them based on the result. The only assumption we make about the distribution of new experimental conditions is that they are drawn from the same pool used in the original study. Specifically, the datasets are selected based on the criteria stated by the authors: they support binary classification, have been previously used in the literature, are available on OpenML, and contain categorical features. These  are the only criteria defining the distribution of datasets in our analysis.
> > > > >
> > > > > - **Section 5.2. I'm confused by n∗ vs. nN∗? What are the different ways they are being predicted?**
> > > > >
> > > > > We thank the reviewer for the question and are happy to clarify the notation used in Sections 5.2 and 5.3. The symbol $n^\*$ refers to the true generalizability threshold, as defined by Equation (3). This is the smallest sample size such that the generalizability exceeds a chosen confidence level $\alpha^\*$ for a given similarity threshold $\varepsilon^\*$. In theory, this value assumes access to all of the possible experiments and can be thought of as $n^\* = n^\*_\infty$. In contrast, $n^\*_N$ is an estimate of $n^\*$ computed from a finite number $N$ of experiments using Algorithm 1. In Section 5.3, because we are working with synthetic data, we are able to compute $n^\*$ directly using its definition by sampling as many experiments as needed. We then compare this to $n^\*_N$, which is computed from subsets of size $N$ to evaluate the quality of the estimate. We clarified these points in the manuscript and changed the notation from $n^\*_N$ to $\hat n^\*_N$ to highlight that it is in fact an estimator.
> > > > >
> > > > > - **"For (B), on the other hand, we estimate n∗=1": the best 2-shot LLM for the observed subtasks is always PALM 535B. Hence the result of a single experiment is enough to acheive (0.95, 0.05)-generalizability". Can the authors explain a little more here? Is the idea: there are T subtasks, all of which PALM performs best on, which would be extremely unlikely to happen by chance? In this case, shouldn't T be an important variable to note?**
> > > > >
> > > > > We thank the reviewer for the insightful question. The interpretation suggested is correct if each LLM had equal probability of being the best on any given subtask. In this case, observing that PALM 535B consistently outperforms all others across $T$ subtasks (with $T = 21$ in this case) would indeed be extremely unlikely, and $T$ would play a central role in quantifying that improbability.
> > > > >
> > > > > However, our analysis proposes a different perspective. Rather than assuming a uniform distribution, we consider the more plausible scenario, supported by empirical evidence, where the LLMs do not have equal chances of outperforming each other. In particular, PALM 535B appears to dominate across all observed subtasks. This suggests that the underlying distribution of results is sharply concentrated: regardless of which experimental condition is sampled, the goal-relevant outcome (e.g., which model is ranked best) remains unchanged.
> > > > >
> > > > > This concentration implies that generalizability can be achieved even with very small sample sizes; in this extreme case, a single experiment is sufficient. However, the challenge lies in validating whether our understanding of the result distribution is correct. That is precisely the motivation for the next section. To clarify this point, we made changes to Section 5.2.

---

### Review · Reviewer_6Gso · 2025-05-15

**Summary Of Contributions:**

This paper focuses on the relatively understudied but important problem of generalizability in experimental studies (especially in ML) – determining if results will hold up when conditions change slightly (e.g., new datasets). Recognizing that many existing methods, such as those based on causal inference, are insufficient for capturing these nuances, the authors provide a mathematical formalization of this problem. When the experimental results are formalized in terms of rankings, the authors propose a statistic that can determine/quantify whether the results are generalizable and develop an algorithm to estimate the appropriate study size for achieving generalizable findings. They demonstrate their approach by analyzing two recent publications, assessing the generalizability of their results, and have released a Python module ("genexpy") for wider use.

**Audience:**

Yes

**Claims And Evidence:**

Yes

**Requested Changes:**

See points above. Also, the definition of n-Generalizability is hard to interpret as $X_j, Y_j$ are not properly defined, and the distance $d$ is represented between random variables rather than their distributions. This conflicts with the definition in the text. This should be clarified through a more precise formulation to avoid confusion.

**Strengths And Weaknesses:**

Strengths:
1. The authors tackle a relatively understudied yet crucial problem and frame it effectively.
2. The paper is well written, and the regular examples make it easy to understand.
3. The algorithm is simple to implement, and the method of finding the number of experiments is interesting.

Weakness:
1. At a high level, the way the problem is formulated, the core of the approach reduces to two-sample testing for ranked data – a problem previously addressed with similar test statistics, for example [1, 2]. The MMD statistic appears exactly as in [2, Eq.6]. I guess the connection should be emphasized in the paper.

2. The relationship between the kernel choice $\kappa$ and the distance $d$ used to define n-generalizability is unclear. It’s not clear whether every distance (or class of distances like f-divergence) has an associated kernel, or if $d(X,Y) = MMD(X,Y)$ by default as in Appendix B; this requires clarification.

3. Can rankings based on scores simply be converted into pairwise comparisons and treated as a two-sample testing problem as in [1]?

4. The methods only work when the number of alternatives is finite. And when the number of alternatives is large, the method may yield overly conservative estimates for the required number of experiments.

5. The way $n*$ is estimated, it becomes a stopping time that is dependent on the data so far. I do not believe definition 4.1 will hold for the estimated $n^*$ as there is dependency between the variables now (but I agree it is a good approximation). In practice, sampling new experiments from the same underlying distribution might not be possible unless they are fixed beforehand (an issue that can be highlighted in the paper).

6. Reporting performance only on a subset of datasets (e.g., best 5) creates potential for manipulation (not something I would consider a weakness of the approach). As it would violate the iid assumption of the Experimental conditions assumed in this paper.

[1] Rastogi, Charvi, et al. "Two-Sample Testing on Ranked Preference Data and the Role of Modeling Assumptions." arXiv preprint arXiv:2006.11909 (2020).
[2] Mania, H., Ramdas, A., Wainwright, M. J., Jordan, M. I., & Recht, B. (2018). On kernel methods for covariates that are rankings.

---

> ### Author Response · Authors · 2025-05-26
>
> We wish to thank the reviewer for their useful feedback and comments. We have revised the manuscript, marking in _green_ the related changes.
>
> - **At a high level, the way the problem is formulated, the core of the approach reduces to two-sample testing for ranked data – a problem previously addressed with similar test statistics, for example [1, 2]. The MMD statistic appears exactly as in [2, Eq.6]. I guess the connection should be emphasized in the paper.**
>
> We thank the reviewer for highlighting these relevant works. We agree that our approach shares similarities with two-sample testing for ranked data, particularly in the use of the MMD statistic. While our method differs in how we utilize the resulting distribution, the underlying statistical foundation is closely related. We have updated the manuscript to reflect this connection more explicitly: we now cite Mania et al. for the definition of the MMD statistic and have included a reference to [1] in Section 4.3 to acknowledge this prior work.
>
> - **The relationship between the kernel choice κ and the distance d used to define n-generalizability is unclear. It’s not clear whether every distance (or class of distances like f-divergence) has an associated kernel, or if d(X,Y)=MMD(X,Y) by default as in Appendix B; this requires clarification.**
>
> We thank the reviewer for this observation. To clarify, our instantiation of $n$-generalizability in Section 4 is specifically based on the Maximum Mean Discrepancy (MMD), and not every distance measure admits  kernels. We have added an explicit explanation in Section 4.2 to make this connection and our choice of distance function clearer.
>
> - **Can rankings based on scores simply be converted into pairwise comparisons and treated as a two-sample testing problem as in [1]?**
>
> Yes, rankings can be converted into pairwise comparisons and treated as a two-sample testing problem as in [1], but there are important distinctions to consider. As discussed in Section 4, rankings are one possible instantiation of the result space in our framework, which remains applicable to other result types as long as suitable kernels can be defined. Even when focusing solely on rankings, the two approaches are not equivalent for two key reasons.
> First, the method in [1] does not incorporate the notion of study-specific goals—an essential motivation for using MMD in our framework. Goals allow experimenters to emphasize specific aspects of the results, and our kernel-based approach explicitly models this flexibility.
> Second, the mathematical formulations differ in expressive power. The approach in [1] represents distributions over preferences using probability matrices, where the entry at $(i,j)$ denotes the probability that alternative $a_i$ beats alternative $a_j$ in a given sample of $l$ rankings. The probability matrix representation is less expressive than distributions over rankings. Consider two distributions of rankings. The first one maps $(0, 1, 2) \mapsto 0.5, (2, 1, 0) \mapsto 0.5$, the second one $(1, 0, 2) \mapsto 0.5, (1, 2, 0) \mapsto 0.5$. The two distributions have the same probability matrix, with $1$ on the diagonal and $0.5$ everywhere else. That said, the method in [1] does allow for more general preference data, while our current work assumes total and transitive relations.
>
> - **The methods only work when the number of alternatives is finite. And when the number of alternatives is large, the method may yield overly conservative estimates for the required number of experiments.**
>
>  We acknowledge that our methods assume a finite number of alternatives, which we consider a reasonable constraint given the nature of experimental studies. Handling infinite or unbounded orderings introduces complexities that are often ill-defined—for example, determining the rank of an element when the ordering extends indefinitely.
> Regarding large numbers of alternatives, it is indeed intuitive that more objects require more experiments to achieve reliable benchmarking. While it is possible that our method may yield conservative estimates in such cases, this behavior depends on the choice of kernel. Characteristic kernels like Mallows, which consider the entire ranking structure, are less likely to be overly conservative, whereas kernels like Jaccard may exhibit greater sensitivity as the number of alternatives grows, since the probability of a specific alternative being ranked best naturally decreases.
> We also note that these challenges are not unique to our approach but are common to related testing methodologies, such as the Friedman test.

---

> > ### Author Response · Authors · 2025-05-26
> >
> > - **The way $n^∗$ is estimated, it becomes a stopping time that is dependent on the data so far. I do not believe definition 4.1 will hold for the estimated $n^∗$ as there is dependency between the variables now (but I agree it is a good approximation). In practice, sampling new experiments from the same underlying distribution might not be possible unless they are fixed beforehand (an issue that can be highlighted in the paper).**
> >
> > In our experiments reported in Section 5, we consistently resample from scratch to ensure independence between samples. To address the reviewer’s concern, we have included additional experiments in the Appendix where resampling is not performed, allowing comparison of results and assessment of potential differences.
> > We agree that, in practice, truly random sampling of new experiments often relies on having a fixed pool of experimental conditions available beforehand. We believe this reflects the reality of most experimental studies, where the experimenter defines the scope and goals upfront and selects appropriate levels for the factors accordingly. We have added a discussion to Section 4.4 to highlight this practical consideration.
> >
> > - **Reporting performance only on a subset of datasets (e.g., best 5) creates potential for manipulation (not something I would consider a weakness of the approach). As it would violate the iid assumption of the Experimental conditions assumed in this paper.**
> >
> > We agree with the reviewer that selectively reporting results on a subset of datasets introduces the potential for manipulation, which would indeed violate the i.i.d. assumption underlying our framework. While such practices are unquestionably problematic, our current method is not designed to detect or mitigate them directly. That said, our approach might be adapted to identify inconsistencies across studies—for instance, if results from similar studies yield significantly different result distributions, this could suggest selective reporting or other irregularities. We have added a note to this effect in Section 6 and consider this a promising direction for future work.
> >
> > - **Also, the definition of n-Generalizability is hard to interpret as $X_j,Y_j$ are not properly defined, and the distance d is represented between random variables rather than their distributions. This conflicts with the definition in the text. This should be clarified through a more precise formulation to avoid confusion.**
> >
> > Thank you for pointing this out. We have revised Section 4.1 to provide a clearer and more precise formulation of $n$-generalizability.

---

> > > ### Comment · Reviewer_6Gso · 2025-05-29
> > >
> > > Thank you for addressing my comments. I have read the authors' responses and found them satisfactory. I have no further questions at this time.

---

### Review · Reviewer_fh8e · 2025-05-15

**Summary Of Contributions:**

The paper investigates the problem of how to ensure that experimental evaluation is sufficient to ensure the method will work in other but similar conditions and on similar problems. The key idea behind the method is to treat parameters of the experiment (parameter can be a dataset) as samples from some distribution. With that, the method suggest to measure the distance of distribution of results from two sets of samples from the parameters of the experiments. The distance is measured by MMD. If those distribution are close, the method will generalize, if they are far, the method will not generalize.

**Audience:**

Yes

**Broader Impact Concerns:**

I think the method might be very useful in practice.

**Claims And Evidence:**

Yes

**Requested Changes:**

I would like authors to discuss my concern with generalization with respect to the datasets.

**Strengths And Weaknesses:**

* I think that the topic of the paper is sound, especially with issues of reproducible science.
* It does not address all concerns. As big part of the reproducibility is that the method is not described with sufficient detail and official implementation sometimes diverge from the description in the paper.
* I think that the main problem of the generalization is the effect of dataset. As I understand, the method assume the distribution of datasets to be similar to what is expected in the practice. While this makes sense from statistical perspective, it is not always true, I do not know, if the proposed method can help with this issue. How one can measure similarity of distributions over datasets?

---

> ### Author Response · Authors · 2025-05-26
>
> We wish to thank the reviewer for their useful feedback and comments. We have revised the manuscript, marking in _blue_ the related changes.
>
> - **I think that the topic of the paper is sound, especially with issues of reproducible science.**
>
> We thank the reviewer for their positive feedback and fully agree that reproducibility is a critical issue in scientific research. In this work, we focus on generalizability, which, while closely related, refers to a distinct concept. Reproducibility concerns the ability to exactly replicate a study under the same conditions, whereas generalizability addresses whether the results hold under new, independently sampled conditions. We emphasize this distinction throughout the paper to avoid confusion and to highlight the complementary roles both concepts play in ensuring robust and reliable scientific findings.
>
> - **It does not address all concerns. As big part of the reproducibility is that the method is not described with sufficient detail and official implementation sometimes diverge from the description in the paper.**
>
> We appreciate the reviewer’s emphasis on the importance of reproducibility, particularly regarding the clarity of methodological descriptions and the alignment between published descriptions and official implementations. As noted in the second paragraph of the Introduction and the citations therein, reproducibility is indeed essential for scientific rigor. However, our work specifically focuses on generalizability, which, while equally important, is a distinct concept. Reproducibility pertains to the ability to exactly replicate a study using the same data and methods, whereas generalizability concerns the consistency of findings under varying experimental conditions. A study may excel in one dimension while lacking in the other—for example, being reproducible but not generalizable if tested on a narrow range of datasets, or vice versa if methods are not publicly accessible. To clarify this distinction, we have revised the opening sentence of the relevant paragraph accordingly.
>
> - **I think that the main problem of the generalization is the effect of dataset. As I understand, the method assume the distribution of datasets to be similar to what is expected in the practice. While this makes sense from statistical perspective, it is not always true, I do not know, if the proposed method can help with this issue. How one can measure similarity of distributions over datasets?**
>
> We thank the reviewer for raising this important point. We agree that the choice and nature of datasets are central to assessing generalizability. In our current work, we do assume that datasets are drawn from a common distribution; however, this assumption is intentionally broad. For example, in our first case study, based on the work of Matteucci et al. (2023), the only criterion for dataset selection was that they support binary classification. In this context, the underlying distribution is loosely defined as “binary classification datasets,” reflecting the practical and often informal criteria used in many empirical studies.
> We acknowledge that measuring similarity between dataset distributions is a complex and open challenge. In the Future Work section, we have revised the first sentence to better reflect our intention to explore methods that incorporate metadata or characteristics of experimental conditions, such as dataset properties, into our generalizability analysis.

---

> > ### Comment · Reviewer_fh8e · 2025-05-26
> >
> > I acknowledge reading the answer. I do not have more questions.

---

### Comment · Action_Editor_32t6 · 2025-05-16
**Start of author discussion**

Dear authors,

The three reviews are now available. Please read them carefully and start discussion. The goal of this period is to gather information so that reviewers can make a convincing decision.

Best,

AE

---

> ### Comment · Action_Editor_32t6 · 2025-05-26
>
> Hi authors,
>
> Please address the concerns raised by the reviewers.
>
> Best,
> AE

---

> > ### Author Response · Authors · 2025-05-26
> >
> > Dear AE,
> >
> > Thank you for the reminder and sorry got eventual delays. We wanted to make sure to properly address all of the concerns raised by the reviewers.
> >
> > Best,
> > the authors

---

### Decision · Action_Editor_32t6 · 2025-06-28

**Recommendation:** Reject

**Audience:**

Yes

**Audience Explanation:**

A general framework for experimental design is surely of broad interest. All reviewers are interested in the paper. One of the reviewers is particularly interested.

**Claims And Evidence:**

No

**Claims Explanation:**

This paper considers generalizability, which ensures that the experiment can be reliably reproduced across multiple runs. The paper introduced a measure-theoretic framework of generalizability based on a similarity measure and rankers. This paper adopts MMD as a similarity measure.

All reviewers agree that the problem is important and undervalued in the community.
Two of the reviewers are positive, whereas the last reviewer is negative on the paper.
* Reviewer fh8e emphasized the importance of the framework, while the reviewer mentioned some points are not addressed. Also, the train-test data shift may be the issue.
* Reviewer 6Gso is also convinced of the importance of the problem. The main concern is that the approach reduces to two-sample testing for ranked data.
* Reviewer CgtL agrees that generalizability in ML is an important problem. The reviewer considers that the exposition can be improved. The reviewer is wondering about the advantage of this general framework over considering each problem setting based on each off-the-shelf statistical testing framework.

After the revision, Reviewer CgtL's main concerns are about (a) the rigorousness of the measure-theoretic framework, and (b) the merit of Equation 4 in 4.4 for estimating the necessary number of experiments, as well as the value of considering such a general framework.

I also had read the paper.
While I am not fully sure about the discussion of Equation 4, I feel it is a good approximation.
* Overall, I agree with Reviewer CgtL in the sense that the paper's presentation can be improved.
* The research question is a tuple Q = (A, C, Igen, X , goals), which is defined at Def 4.1. The paper may be clearer by explaining how the elements in examples (e.g., Example 4.1) correspond to the question tuple.
* Maybe I am mistaken, but the distance metric is not included in the tuple. The "goal" in the tuple can be more well-defined.
In the introduction of this paper, it compares the generalizability (of this paper) with a relevant notion of replicability. In my view, replicability is more solidly defined. It is essentially a property of an algorithm under some data-generating process (DGP) such that two different datasets from the DGP replicate the results. I think authors can define ($\delta$, $\varepsilon$)-generalizability as a property of an experimental process by using a Definition clause.
* I think the metric on experimental results (Section 4.3) is necessary since it is essential in the definition of ($\delta$, $\varepsilon$)-generalizability, whereas I am not sure how important the ranking is on the framework in the paper. I am not sure the goal, ranking, metric are clearly defined in the research question tuple above.

Minor:

* I am not sure whether the bullet points in "experimental factors" in Section 3.1 are rigorously defined. This is somewhat descriptive rather than mathematical setup.

In summary, all reviewers are interested in the paper, and at least two of the reviewers are convinced of the problem setup (one is excited), while several concerns remain. Based on these discussions, I recommend rejecting with a recommendation for resubmission after a major revision.

**Resubmission Of Major Revision:**

The authors may consider submitting a major revision at a later time.